# An automated feeding system for the African killifish reveals the impact of diet on lifespan and allows scalable assessment of associative learning

Andrew McKay[1,2†], Emma K Costa[3,4†], Jingxun Chen[1†], Chi-Kuo Hu[1], Xiaoshan Chen[1], Claire N Bedbrook[1,5], Rishad C Khondker[1], Mike Thielvoldt[6], Param Priya Singh[1], Tony Wyss-Coray[3,7,8], Anne Brunet[1,7,8*]

[1]Department of Genetics, Stanford University, Stanford, United States; [2]Biology Graduate Program, Stanford University, Stanford, United States; [3]Department of Neurology and Neurological Sciences, Stanford University, Stanford, United States; [4]Neurosciences Interdepartmental Program, Stanford University School of Medicine, Stanford, United States; [5]Department of Bioengineering, Stanford University, Stanford, United States; [6]Thielvoldt Engineering, Albany, United States; [7]Glenn Laboratories for the Biology of Aging, Stanford University, Stanford, United States; [8]Wu Tsai Neurosciences Institute, Stanford University, Stanford, United States

*For correspondence:
abrunet1@stanford.edu

†These authors contributed equally to this work

Competing interest: The authors declare that no competing interests exist.

**Abstract** The African turquoise killifish is an exciting new vertebrate model for aging studies. A significant challenge for any model organism is the control over its diet in space and time. To address this challenge, we created an automated and networked fish feeding system. Our automated feeder is designed to be open-source, easily transferable, and built from widely available components. Compared to manual feeding, our automated system is highly precise and flexible. As a proof of concept for the feeding flexibility of these automated feeders, we define a favorable regimen for growth and fertility for the African killifish and a dietary restriction regimen where both feeding time and quantity are reduced. We show that this dietary restriction regimen extends lifespan in males (but not in females) and impacts the transcriptomes of killifish livers in a sex-specific manner. Moreover, combining our automated feeding system with a video camera, we establish a quantitative associative learning assay to provide an integrative measure of cognitive performance for the killifish. The ability to precisely control food delivery in the killifish opens new areas to assess lifespan and cognitive behavior dynamics and to screen for dietary interventions and drugs in a scalable manner previously impossible with traditional vertebrate model organisms.

## Editor's evaluation

The manuscript by McKay et al., describes in detail a novel feeding system for killifish that allows high precision control of feeding amount and schedule on a per-tank basis. The system is open-source, utilizing 3D-printed and off-the-shelf components and due to this emphasis on open software and hardware, the system can be built by individual research groups and the approach therefore appears highly scalable. The authors demonstrate the value of precise control of food intake by investigating sex-specific lifespan effects and associated transcriptome responses to DR in killifish. The resources described in this paper will be a major step in killifish husbandry and management to facilitate its use as a model of longevity studies.

## Introduction

The African turquoise killifish, *Nothobranchius furzeri*, is a new genetically tractable model organism that has been developed for the study of aging and 'suspended animation' (embryonic diapause) (*Cellerino et al., 2016*; *Harel and Brunet, 2015*; *Hu and Brunet, 2018*; *Platzer and Englert, 2016*; *Poeschla and Valenzano, 2020*). This fish has a naturally compressed lifespan of 4–6 months, which is 6 times shorter than the maximum lifespan of mice and 10 times shorter than zebrafish (*Harel and Brunet, 2015*). In its short life, the African killifish exhibit hallmarks of aging, including cognitive decline (*Valenzano et al., 2006a*; *Valenzano et al., 2006b*), neurodegeneration (*Bagnoli et al., 2022*; *Matsui et al., 2019*; *Terzibasi et al., 2008*), fertility decline (*Api et al., 2018*; *Zak and Reichard, 2021*), cellular senescence (*Valenzano et al., 2006a*), impaired regeneration and wound healing (*Wendler et al., 2015*), defects in heart function (*Ahuja et al., 2019*), and an increased risk of cancer (*Baumgart et al., 2015*; *Di Cicco et al., 2011*). A genetic and genomic toolkit has been developed for the killifish, including sequencing of its genome (*Reichwald et al., 2015*; *Valenzano et al., 2015*), Tol2-based transgenesis (*Hartmann and Englert, 2012*; *Valenzano et al., 2011*), and CRISPR/Cas9-mediated genome-editing (*Harel et al., 2015*). These developments have allowed disease modeling in the killifish (*Harel et al., 2015*), identification of genes potentially involved in lifespan differences (*Reichwald et al., 2015*; *Valenzano et al., 2015*) and sex determination (*Reichwald et al., 2015*; *Valenzano et al., 2009*), determination of enhancers involved in tissue regeneration (*Wang et al., 2020*), and discovery of chromatin regulators important for embryonic diapause (*Hu et al., 2020*).

A major challenge for any model system is the precise control of its diet. A proper diet is important for robust growth and fertility in colony maintenance and genetic manipulations. A controlled feeding regimen is also critical for lifespan studies, given the impact of dietary restriction – and more generally diet – on lifespan in a wide variety of species (*Fontana and Partridge, 2015*; *Green et al., 2022*; *Longo and Anderson, 2022*; *Mair and Dillin, 2008*), including worms (*Houthoofd and Vanfleteren, 2007*), flies (*Partridge et al., 2005*), killifish (*Terzibasi et al., 2009*), stickleback fish (*Inness and Metcalfe, 2008*), mice (*Bartke et al., 2001*; *Mitchell et al., 2019*; *Weindruch et al., 1986*), rats (*Goodrick et al., 1983*; *Swindell, 2012*; *Turturro et al., 1999*), and monkeys (*Colman et al., 2009*; *Colman et al., 2014*; *Mattison et al., 2017*; *Mattison et al., 2012*). Importantly, food is not only essential for growth, fertility, and survival, but it can also serve as a reward. Indeed, several cognitive tests rely on the ability of associating food with a task (*Flagel and Robinson, 2017*; *Jarrard, 1993*; *Meyer et al., 2012*; *Olton and Samuelson, 1976*; *Rudy and Sutherland, 1989*).

In the wild, adult African turquoise killifish primarily feed upon small crustaceans and insect larvae ('bloodworms') that are present in their ephemeral ponds in Africa (*Polacik and Reichard, 2010*; *Reichard and Polačik, 2019*). In laboratory settings, African killifish are often fed using bloodworms, either live (*Hartmann et al., 2011*; *Reichwald et al., 2015*; *Terzibasi et al., 2009*; *Wendler et al., 2015*), frozen (*Genade et al., 2005*; *Terzibasi et al., 2008*; *Valenzano et al., 2006b*; *Zupkovitz et al., 2018*), or lyophilized (*Harel et al., 2015*; *Hu et al., 2020*; *Valenzano et al., 2015*). However, feeding with live bloodworms can introduce pathogens in the colony (*Broza and Halpern, 2001*; *Moore et al., 2003*; *Rouf and Rigney, 1993*), and the nutrition values of live bloodworms can vary depending on the lot and supplier (*Fard et al., 2014*). To alleviate these issues, dry food pellets have been recently adopted for killifish, either alone (*Zak et al., 2020*; *Zak et al., 2022*) or in combination with live food (*Matsui et al., 2019*). But a main challenge of dry fish food – and fish food in general – is that it needs to be delivered for each feeding, otherwise it loses its appeal to fish and remains uneaten in the tanks. Regardless of the type of food used in the laboratory, feeding fish is largely based on manual feeding. Hence, food delivery is a limiting factor: it is hard to perform in a consistent manner, to scale up, and to schedule at any time of the day or night. These features hamper the testing of different feeding regimens and other interventions. Control over food delivery will help the development of the killifish as a scalable model for lifespan and other traits and to allow interventions.

To address these challenges and develop the scalability of the killifish as a model system, we have created an automated feeding system for killifish feeding. We provide evidence that this system is precise and reliable, and that it allows controlled and tunable feeding throughout the day or night. Using this new flexible feeding system, we explore the parameters of diet in the killifish and identify a dietary restriction regimen that extends the lifespan of killifish and modulates liver transcriptome in a sex-specific manner. Interestingly, our automated feeders also allow us to design a novel associative learning assay to test cognitive function in the African killifish. This automated feeding system will

help the development of the killifish as a high-throughput model for lifespan and will allow scalable intervention or drug screening.

## Results

### A wireless networked automated feeding system

Controlling feeding automatically is a critical component for the development and scalability of a model organism. Automated feeders have been developed for fish, but they are rarely used due to their imprecision (e.g., hobbyist feeding systems) or prohibitive costs (e.g., scientific-grade feeding systems, such as Tritone from Tecniplast). While systems developed for zebrafish have solved some of these constraints (*Doyle et al., 2017*; *Lange et al., 2021*), they are either not scalable to hundreds of animals simultaneously being fed or cannot easily be added onto commercial water systems. Thus, there is still a need for high-precision and programmable feeders compatible with commonly used water systems.

To address the main limitations with current feeding methods, we developed a wireless networked automated feeding system for the African turquoise killifish. We created a system in which different components function independently to confer robustness and avoid single points of failure that affect the overall feeding scheme. This is particularly important when feeding needs to happen over a lifespan. The automated feeder that we designed and built is placed on top of each animal's 2.8 L tank (*Figure 1A*) and drops dry food (e.g., Otohime fish diet) from a small feed hopper (*Figure 1B*) directly into the tank (which houses one individual fish). The feeder is powered by an attached battery, and the food pellets are automatically segregated from the hopper by a rotating acrylic disc, with the resulting pieces of food dropping onto the water through a 3-mm diameter opening cut out in the supporting acrylic plate below (*Figure 1C*, *Figure 1—figure supplement 1A*, *Figure 1—video 1*). The 3-mm opening rotates from under the feed hopper, collects food, and travels to the drop site above the tank opening. Each rotation delivers a fixed volume of food, averaging around 5 mg in mass, and multiples of 5 mg can be programmed to increase food amount per feeding. Feedings are also fully programmable to any frequency or time of day by the user, and each feeder operates independently of one another (*Figure 1—figure supplement 1B*), allowing flexibility in feeding schedule (*Table 1*).

To determine whether food has indeed dropped into the tank, we designed the acrylic disc's opening such that it pushes the food between a photoresistor and a light-emitting diode (LED) before reaching the drop site. The photoresistor and LED provide confirmation for feeding by measuring the resistance of the photoresistor when food obstructs the light from the LED (outgoing trip), and after feeding, when the empty food-receptacle 3-mm opening allows light to pass (return trip) (*Figure 1—figure supplement 1A*, steps 3–6). We also designed the feeder such that each feeder communicates feeding confirmations independently to a local server using the 802.11 wireless communication standard, which can then be aggregated across groups of feeders to a cloud-based server (*Figure 1D*). Thus, with our automated system, feedings can be recorded and backed up remotely, providing an automatic log for the user.

Lastly, the wireless communication and battery-powered function allows our system to function remotely and flexibly. This is an improvement over other automated systems (such as Tritone from Tecniplast), whose monolithic design creates many single points of system-wide failure, or over designs that are not networked or restricted to less flexible wire-based communication (*Doyle et al., 2017*; *Manabe et al., 2013*; *Yang et al., 2019*). We found that more than 100 automated feeders can operate simultaneously on the same local server. Additionally, our automated feeders are estimated to cost 12.88 USD per feeder (*Figure 1—source data 1*), so ~2000 USD for >100 tanks (including network setup and operation cost for 2 years), which is significantly less expensive than commercial systems (e.g., the Tecniplast Tritone system costs ~200,000 USD to feed approximately 240 tanks). Hence, our design for an automated feeding system allows controlled, tunable, and recorded feedings throughout the lifespan of the killifish.

### Fidelity and precision of the automated feeding system

We tested the fidelity and precision of the automated feeding system. A representative feeder was set to deliver 7 feedings a day for 30 days, and food delivery was recorded on the server (*Figure 1—figure supplement 1A*, steps 3–6). Food delivery occurred 98.1% of the time (206 actual food delivery for

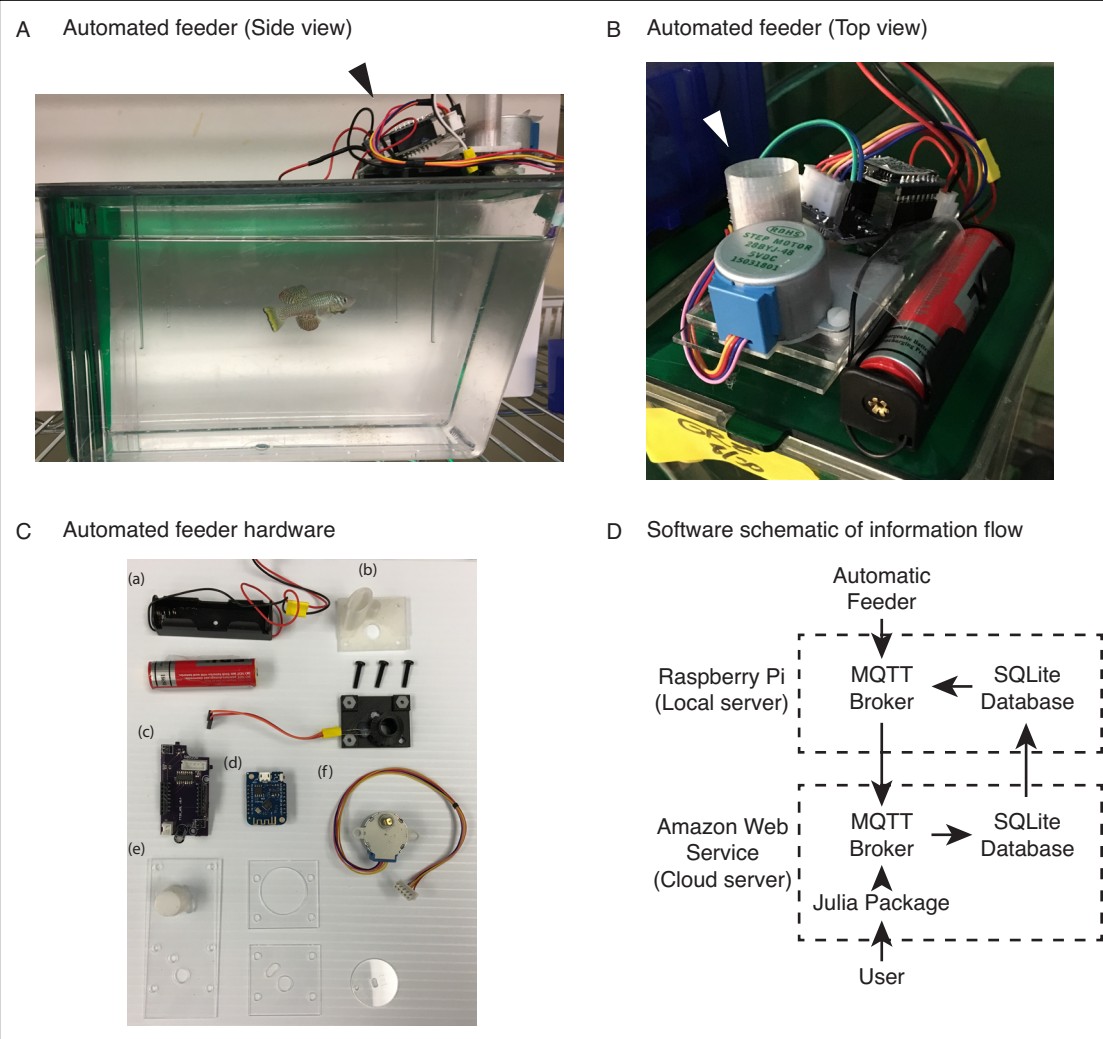

**Figure 1.** An automated 3D-printed feeding system for the African turquoise killifish. (**A**) Side view of an individual automated feeding unit placed on the top front of a 2.8 L fish tank supplied by Aquaneering. At the midpoint of the tank height, a 2.8 L tank is approximately 23 cm long from front to back. Dark arrowhead: feeding unit on the lid of a tank. (**B**) Top view of an individual automated feeding unit. This is a compact, self-contained unit with individual power supply, food hopper, stepper motor for food delivery, and microcontroller for control and communication. White arrowhead: hopper where the dry food is placed. (**C**) Components of an individual automated feeder. Each feeder is composed of a lithium ion battery and holder (**a**), 3D-printed parts (**b**) coupled with a custom-printed circuit board (**c**), a Wemos D1 mini ESP8266-based microcontroller board (**d**), laser-cut acrylic parts (**e**), and stepper motor (**f**). (**D**) Users control the automated feeding system by interacting with a cloud-based server. This server communicates with small local servers on the premise, which then communicate with the individual automated feeders. Changes to feeding schedules are provided by users and filter down to the appropriate feeders, while status updates, such as feeding confirmations, make the return trip back to the users.

The online version of this article includes the following video, source data, and figure supplement(s) for figure 1:

**Source data 1.** Automatic feeder parts list.

**Figure supplement 1.** Automated feeders are designed to confirm each feeding and can be programmed to perform independent feeding regimens.

**Figure 1—video 1.** Example of automatic feeder function.

https://elifesciences.org/articles/69008/figures#fig1video1

210 scheduled food deliveries; *Figure 2A*). Overall, aggregating 41 feeders for 2279 cumulative days showed that most food deliveries were fully accounted for (*Figure 2B*), with only 7.89% of days deviating by one unconfirmed food delivery (*Figure 2C*). This fidelity was confirmed independently with a separate set of automated feeders of the same design built independently by another researcher (*Figure 2—figure supplement 1A and B*).

We compared the precision of the mass of food dropped per delivery for our automated feeder versus manual feeding by individual users. Single or multiple automated feeders were more precise,

**Table 1.** Automated feeding allows for more flexible, frequent, and precise feedings than manual feeding.

Manual feeding is restricted to the times of the day when researchers can enter the animal facilities based on day/night cycles, whereas automated feeding can be programmed to occur at any time of the day or night. Manual feeding is restricted by the researcher's ability to feed, while automated feeding can occur as frequently as 144 times per day. Manual feeding requires individual attention to each tank on a daily basis, multiple times per day, whereas automated feeding requires an initial setup and biweekly checks to replace batteries and add food to the devices. Manual feeding is relatively imprecise at 0.016 (the inverse of the variance of food delivered), while automated feeding has a higher precision at 0.512.

|  | **Manual feeding** | **Automated feeding** |
|---|---|---|
| Schedule flexibility | Limited to weekday working hours | 24/7 |
| Maximum frequency | 2–3 times a day | 144 times a day |
| Labor required | Daily, linearly increasing with tanks | Initial setup and biweekly checks |
| Precision | Low, with batch effects for different individuals (0.016) | Higher (0.512) |

by an order of magnitude, at delivering a given amount of food compared to a group of six different individuals (similar to what can be done in fish rooms to offset the workload) or a single individual measuring and delivering food (*Figure 2D*).

Because of its flexibility, the automated feeder can deliver up to 5 mg per unit every 10 min, or 720 mg per 12 hr period, representing a potential 20× increase over the baseline dietary regime for the day (*Table 1*). The automated feeder can also feed during the night, if desired, which would not be practical for manual feeding. Thus, our automatic system decreases the amount of labor and provides high precision, reproducibility, and flexibility for husbandry and for varying diet regimens.

## Defining a daily dietary restriction feeding schedule in the killifish

We used our automated feeding system to define a variety of dietary regimens in killifish – dietary restriction and overfeeding. Dietary restriction has been shown to delay signs of aging and age-related diseases in multiple species (*Fontana and Partridge, 2015*; *Green et al., 2022*; *Longo and Anderson, 2022*; *Mair and Dillin, 2008*). Dietary restriction regimens encompass restricting overall food (*Colman et al., 2009*; *Colman et al., 2014*; *Mattison et al., 2017*; *Mattison et al., 2012*; *Weindruch et al., 1986*) or restricting the time of feeding during the day (*Mitchell et al., 2019*) or over longer periods (*Brandhorst et al., 2015*). In killifish, dietary restriction has been done by every other day feeding because it is difficult to do otherwise with manual feeding (*Terzibasi et al., 2009*). A dietary restriction regimen is expected to reduce growth and fertility (especially when applied early in adulthood) compared to an ad libitum feeding regimen. In contrast, overfeeding has negative consequences on health, and it is expected to increase growth but to reduce fertility compared to an ad libitum regimen (*Magwere et al., 2004*). To define different dietary regimens in the African turquoise killifish, we fed individually housed male and female killifish – starting in young adults (1 month of age) – and varied both the amount and timing of feedings throughout the day using our programmable automated feeders. For 'ad libitum' (AL, blue), we fed 35 mg of dry food per day, in seven feedings of 5 mg evenly spaced over 12 hr of the day, for a total of 245 mg per week (roughly similar to manual feeding) (*Figure 3A*). For 'dietary restriction' (DR, orange), we fed 15 mg per day (~57% restriction), in three feedings of 5 mg over 2 hr, for a total of 105 mg per week (*Figure 3A*). This DR regimen is both amount- and time-restricted (achieving amount restriction without time restriction would not be possible in current settings because of the minimum 5 mg delivery amount of automated feeder, see above). This DR regimen led to smaller (*Figure 3B and C*) and less fertile (*Figure 3D*, *Figure 3—figure supplement 1A and B*) animals than the AL regimen, consistent with what is expected under dietary restriction. Importantly, at seven feedings of 5 mg a day (AL), animals were not overfed because they could be fed more (twelve feedings of 5 mg a day) (*Figure 3E*), and this extra feeding increased size (*Figure 3F and G*) but decreased fertility (*Figure 3H*, *Figure 3—figure supplement 1C and D*). Hence, a favorable feeding regimen, at least under these husbandry conditions, is around seven

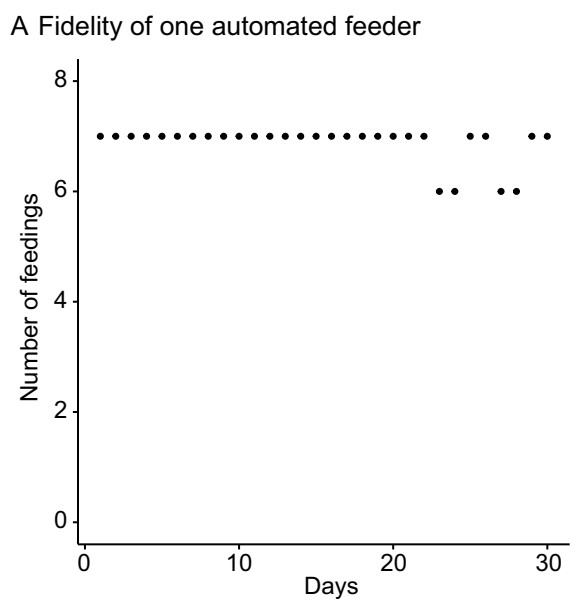

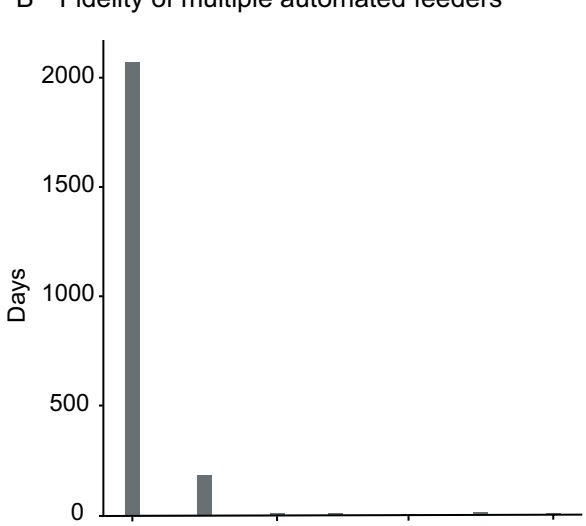

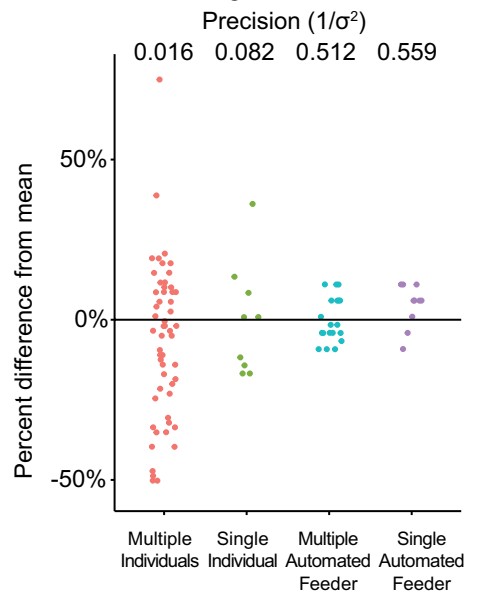

**Figure 2.** Fidelity and precision of the automated feeding system. (**A**) Logged feedings per day from a representative feeder over a 30-day period, with seven feedings of 5 mg programmed for each day. Note that a recording of six feedings instead of seven does not necessarily mean that the feeding was missed but could be due to an unlogged feeding due to inability to connect to main server. Source data: ***Figure 2—source data 1***. (**B, C**) Histogram (**B**) of deviations from scheduled feedings for 2279 days of feedings over 41 feeders with a given number of deviations tabulated in (**C**). The vast majority (>98%) of feedings have zero or one missed feeding. Source data: ***Figure 2—source data 2***. (**D**) Automated feeding provides higher feeding precision than manual feeding as seen from the dispersion of feeding volumes as a percentage of the mean. When compared to manual feeding by multiple individual users or a single individual, automated feeders are more precise in the amount of food delivered. Precision is defined as the reciprocal of the estimated variance given either six individuals (n = 54, precision = 0.016), one individual (n = 9, precision = 0.082), four feeders (n = 19, precision = 0.512), or one single feeder (n = 10, precision = 0.559). Precision derived from bootstrapped estimates of the standard deviation for each group. Source data: ***Figure 2—source data 3***.

The online version of this article includes the following source data and figure supplement(s) for figure 2:

**Source data 1.** Feeding logs from a representative feeder over a 30-day period.

**Source data 2.** Missed feedings for 41 feeders over 2279 days of feeding.

*Figure 2 continued on next page*

*Figure 2 continued*

**Source data 3.** Feeding data for automatic and manual feedings.

**Source data 4.** Missed feedings for 90 feeders over 9284 days of feeding.

**Figure supplement 1.** Fidelity and precision of the automated feeding system validated by independent researcher.

feedings per day and 35 mg of food per day for the African killifish. Together, these experiments define diets that optimize growth and fertility in the African killifish (for these husbandry conditions), and they identify a dietary restriction regimen in this species.

## An amount- and time-restricted diet regimen extends lifespan in a sex-specific manner

We asked whether the amount- and time-restricted DR regimen defined above could promote longevity in the killifish. To this end, we assessed the lifespan of female and male African killifish in ad libitum (AL, blue: seven evenly spaced feedings of 5 mg per day over 12 hr of the day) or the amount- and time-restricted conditions (DR, orange: amount- and time-restricted: three feedings of 5 mg per day over 2 hr in the morning) (*Figure 4A*). We enrolled young adult animals in two independent cohorts (36 males and 32 females in cohort 1 [*Figure 4—figure supplement 1A*], and 42 males and 49 females in cohort 2 [*Figure 4A*]). The DR regimen was initiated in young adults (1 month of age) and lasted until death (*Figure 4A*). Interestingly, males fed the DR diet (orange) lived longer than those fed the AL diet (blue; 16.6% median lifespan extension for cohort 1 [*Figure 4—figure supplement 1B*], 22.1% median lifespan extension for cohort 2 [*Figure 4B*]). In contrast, females fed the DR diet did not live significantly longer than those fed the AL diet in either cohort (*Figure 4C*, *Figure 4—figure supplement 1C*). Animals fed a DR diet exhibited lifespan differences between sexes, with male living significantly longer than females (*Figure 4D and E*, *Figure 4—figure supplement 1D and E*). The sex-specific effect of DR on killifish lifespan was also supported using Cox proportional hazards in a factorial design (*Figure 4—source data 3*), where the interaction term between sex and dietary regimen was found to be significant (p=0.045). Furthermore, fitting the survival data of male killifish into a Gompertz distribution resulted in an estimated reduced slope for males in DR conditions. These results suggest that this DR regimen reduces the 'rate of aging' in male killifish (*Figure 4F*), in line with the effect of intermittent feeding on lifespan (*Terzibasi et al., 2009*). Thus, this DR regimen (restricted in time and amount) significantly extends the lifespan of males, but not females, in the African killifish.

## An amount- and time-restricted diet regimen impacts the liver transcriptome in a sex-specific manner

To determine whether this amount- and time-restricted DR regimen could impact gene expression in a sex-specific manner, we generated transcriptomic datasets. We focused on liver and brain because of their known roles in systemic metabolism and dietary response (*Ye and Medzhitov, 2019*). Using the automated feeders, we initiated DR or AL regimens in females and males at the young adult stage (1 month of age) and fed them these regimens for 5 weeks (*Figure 5A*). We then collected livers and brains for RNA-sequencing (RNA-seq) (*Figure 5A*). Principal component analysis (PCA) on the full transcriptomes showed strong separation by sex in the liver in the AL condition, and this separation was even larger in the DR condition (*Figure 5B*, *Figure 5—figure supplement 1A*). In contrast, PCA did not reveal detectable separation by sex or diet in the brain (*Figure 5B*, *Figure 5—figure supplement 1B*). Thus, liver gene expression is sex specific in AL and DR conditions, and this sexual dimorphism is exacerbated in the DR condition.

We next identified the set of genes that are differentially regulated by diet (AL vs. DR) for each sex. This analysis yielded 221 differentially expressed genes between AL and DR conditions ('diet DEGs') for females and 70 diet DEGs for males (p<0.05), with 10 genes shared between the two sexes (*Figure 5—figure supplement 1C*). Gene set enrichment analysis (GSEA) revealed that fatty acid and acetyl-CoA metabolism genes (e.g., *FADS2*, *ACSS2*, *DGAT2*, *SCD5*) were strongly downregulated in the liver in response to DR in females, but less so in males (*Figure 5C and D*, *Figure 5—source data 6*, *Figure 5—source data 7*). Protein folding and ER stress response pathways (e.g., *HSP90B1*, *DNAJB11*, *SDF2L1*, *DERL1*) were strikingly upregulated in the liver in response to DR in females, but not in males (*Figure 5C and D*). Hypergeometric Gene Ontology (GO) enrichment analysis of the

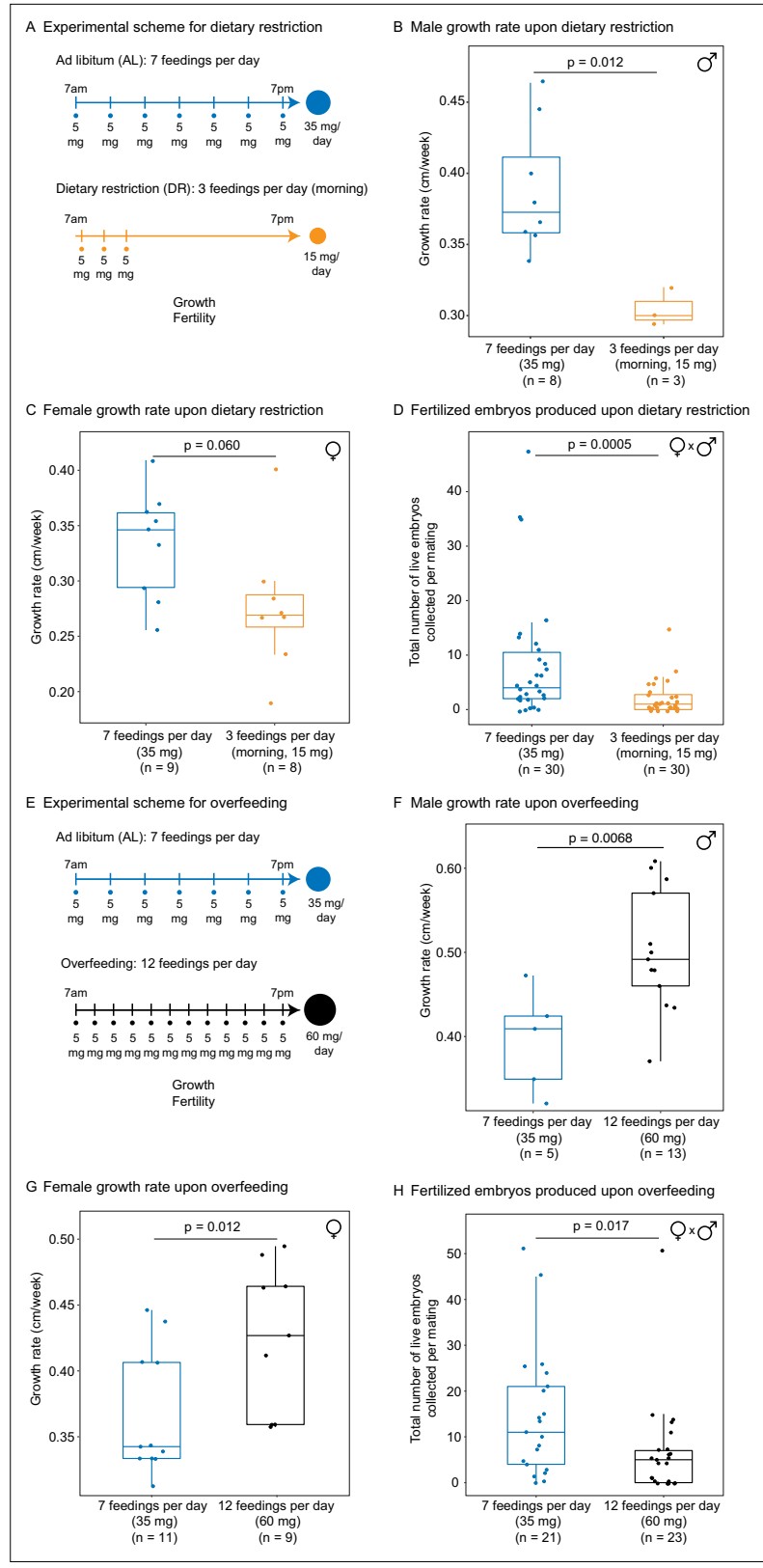

**Figure 3.** Automated feeding enables the definition of different diets for the African killifish. (**A**) Experimental scheme to compare two automated feeding schedules for killifish: feeding seven times a day evenly over 12 hr (5 mg Otohime fish diet per feeding, 35 mg total per day) and feeding three times a day within a 2 hr period in the morning (5 mg Otohime fish diet per feeding, 15 mg total per day). These regimens were applied from

*Figure 3 continued on next page*

*Figure 3 continued*

1 month of age to death (for growth measurements) and from 1 month of age until death of one individual in the pair (for fertility measurements). For fertility measurements, each individual in the pair was fed individually while single-housed and then crossed for 24 hr once per week to assess fertility. (**B**) The average growth rate (cm/week) of male killifish from 1 month of age to death fed either seven times a day (blue, median = 0.3726 cm/week, n = 8) is significantly greater than those fed three times a day in the morning (orange, median = 0.3 cm/week, n = 3). Each dot represents a single animal's growth averaged over its lifespan. Animals that lived longer than 4 months were not considered due to concerns with nonlinear growth rate. Significance determined by Wilcoxon rank-sum test (p=0.012). Animals were from the same cohort as *Figure 4—figure supplement 1* (cohort 1). Source data: *Figure 3—source data 1*. (**C**) The average growth rate (cm/week) of female killifish from 1 month of age to death fed seven times a day (blue, median = 0.3461 cm/week, n = 9) is greater than those fed three times a day in the morning (orange, median = 0.2690 cm/week, n = 8). Each dot represents a single animal's growth averaged over their lifespan. Animals that lived longer than 4 months were not considered due to concerns with nonlinear growth rate. Significance determined by Wilcoxon rank-sum test (p=0.060). Animals were from the same cohort as *Figure 4—figure supplement 1* (cohort 1). Source data: *Figure 3—source data 1*. (**D**) Killifish mating pairs (one male and one female) fed three times a day are significantly less fertile (median = 1 fertilized embryo per mating, n = 30 matings across five pairs) than mating pairs fed seven times a day (median = 4 fertilized embryos per mating, n = 30 matings across five pairs). Each dot represents the fertilized embryos collected from one pair crossing overnight, and pairs were mated from 8 weeks of age until one individual in the pair died. Significance determined by Wilcoxon rank-sum test (p=0.0005). Source data: *Figure 3—source data 2* and *Figure 3—source data 3*. (**E**) Experimental scheme to compare two automated feeding schedules for killifish: feeding twelve times a day (5 mg Otohime fish diet per feeding, 60 mg total per day) compared to feeding seven times a day (5 mg Otohime fish diet per feeding, 35 mg total per day). These regimens were applied from 1 month of age to death (for growth measurements) and from 2 months of age until death of one individual in the pair (for fertility measurements). For fertility measurements, each individual in the pair was fed individually while single-housed and then crossed for 24 hr once per week to assess fertility. (**F**) The average growth rate (cm/week) of male killifish from 1 month of age to death is significantly higher for animals fed twelve times a day (black, median = 0.4917 cm/week, n = 13 males) than for animals fed seven times a day (blue, median = 0.4092 cm/week, n = 5 males). Each dot represents a single animal's growth averaged over its lifespan. Animals that lived longer than 4 months were not considered due to concerns with nonlinear growth rate. Significance determined by Wilcoxon rank-sum test (p=0.0068). Source data: *Figure 3—source data 4*. (**G**) The average growth rate (cm/week) of female killifish from 1 month of age to death is significantly higher for animals fed twelve times a day (black, median = 0.4268 cm/week, n = 9 females) than for animals fed seven times a day (blue, median = 0.3425 cm/week, n = 11 females). Each dot represents a single animal's growth averaged over its lifespan. Animals that lived longer than 4 months were not considered due to concerns with nonlinear growth rate. Significance determined by Wilcoxon rank-sum test (p=0.012). Source data: *Figure 3—source data 4*. (**H**) Killifish mating pair fertility is significantly lower for animals fed twelve times a day (black feeding schedule, median = 5 fertilized embryos per mating, n = 23 matings) than for animals fed seven times a day (blue feeding schedule, median = 11 fertilized embryos per mating, n = 21 matings). Each dot represents fertilized embryos collected from one pair upon crossing for 24 hr and pairs were mated from 2 months of age until one individual in the pair died. Significance determined by Wilcoxon rank-sum test (p=0.017). Source data: *Figure 3—source data 5* and *Figure 3—source data 6*.

The online version of this article includes the following source data and figure supplement(s) for figure 3:

**Source data 1.** Metadata table for growth rate comparisons for males and female killifish on dietary restriction (DR) diet regimens.

**Source data 2.** Embryo viability data for mate pairs on dietary restriction (DR) or ad libitum (AL) diet regimens.

**Source data 3.** Metadata table for mate pairs used in embryo viability experiments in *Figure 3—source data 2*.

**Source data 4.** Metadata table for growth rate comparisons for males and female killifish on OE diet regimens.

**Source data 5.** Embryo viability data for mate pairs on overfeeding or ad libitum (AL) diet regimens.

**Source data 6.** Metadata table for mate pairs used in embryo viability experiments in *Figure 3—source data 5*.

**Figure supplement 1.** Total number of embryos produced in response to different diets.

diet DEGs confirmed both observations (see 'Materials and methods,' *Figure 5—figure supplement 1D*, *Figure 5—source data 8*). Finally, GSEA (but not the hypergeometric GO analysis) showed that genes involved in inflammation-related pathways – positive regulators of immune responses, leukocyte activation, and cytokine-mediated signaling (e.g., *CMKLR1*, *TBK1*, *BCL6*) – were strongly downregulated in response to DR in the liver in males, but less so in females (*Figure 5C and D*). Thus, the

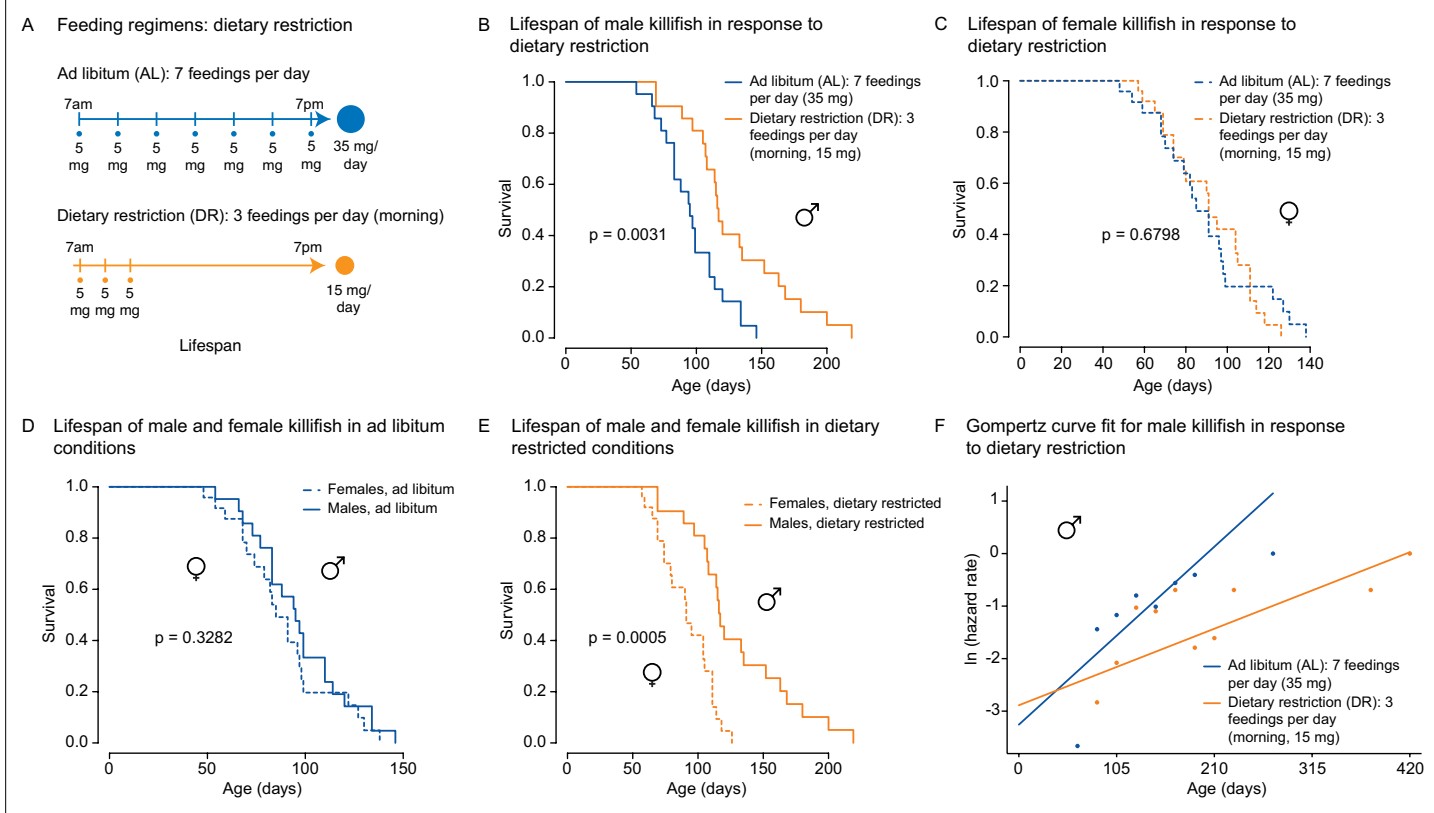

**Figure 4.** An amount- and time-restricted dietary regimen robustly extends lifespan in male, but not female African turquoise killifish. (**A**) Experimental scheme comparing two automated feeding schedules for killifish: an ad libitum (AL) regimen (seven times a day, 35 mg Otohime fish diet per day, blue) or a dietary restricted (DR) regimen (three times in the morning, 15 mg Otohime fish diet per day, orange). Automated feeding regimens were started after sexual maturity, at 1 month of age. (**B**) Male killifish fed a DR regimen (solid orange, median lifespan = 116 days, n = 21) lived significantly longer than male killifish fed an AL regimen (solid blue, median lifespan = 95 days, n = 21) (p=0.0031, log-rank test). Source data: *Figure 4—source data 1*. (**C**) Female killifish fed a DR regimen (dashed orange, median lifespan = 90 days, n = 25) did not live significantly longer than female killifish fed an AL regimen (dashed blue, median lifespan = 82.5 days, n = 24) (p=0.6798, log-rank test). Source data: *Figure 4—source data 1*. (**D**) In AL conditions, male killifish (solid blue, median lifespan = 95 days, n = 21) did not exhibit lifespan difference from female killifish (dashed blue, median lifespan = 82.5 days, n = 24) (p=0.3282, log-rank test). Source data: *Figure 4—source data 1*. (**E**) In DR conditions, male killifish (solid orange, median lifespan = 116 days, n = 21) lived significantly longer than female killifish (dashed orange, median lifespan = 90 days, n = 25) (p=0.0005, log-rank test). Source data: *Figure 4—source data 1*. (**F**) Fitted curve of the binned 27-day hazard rate of male killifish from both cohorts to a Gompertz distribution and then transformed into the natural log of the hazard rate. The estimated 'rate of aging' (slope) of killifish on the AL feeding regimen is 0.3388 (95% confidence interval = 0.2437–0.434) and is significantly different from the rate of aging for DR, which is 0.1457 (95% confidence interval = 0.0846–0.2069). The estimated 'frailty' (intercept) for killifish on the AL feeding regimen is 0.0385 (95% confidence interval = 0.0203–0.073) compared to 0.0557 (95% confidence interval = 0.0315–0.0986) for killifish on DR (overlapping confidence intervals for intercept parameter indicate that these parameters are not significant). Source data: *Figure 4—source data 1* and *Figure 4—source data 2*.

The online version of this article includes the following source data and figure supplement(s) for figure 4:

**Source data 1.** Lifespan data for male and female killifish on ad libitum (AL) and dietary restriction (DR) diet regimens (cohort 2).

**Source data 2.** Lifespan data for male and female killifish on ad libitum (AL) and dietary restriction (DR) diet regimens from independent cohort (cohort 1).

**Source data 3.** Result from proportional hazards model that shows dietary restriction (DR) independently and significantly reduces the hazard rate.

**Figure supplement 1.** An independent cohort of African turquoise killifish undergoing the amount- and time-restricted dietary regimen also exhibits lifespan extension in males.

sex-specific response to diet in the liver involves lipid metabolism, protein homeostasis, and inflammatory signaling pathways.

We also identified the set of genes that are differentially regulated by sex (male vs. females) for each regimen ('sex DEGs'). Interestingly, sex DEGs (in either AL or DR) were significantly enriched in the diet DEGs of either males or females, controlling for gene expression distribution and gene

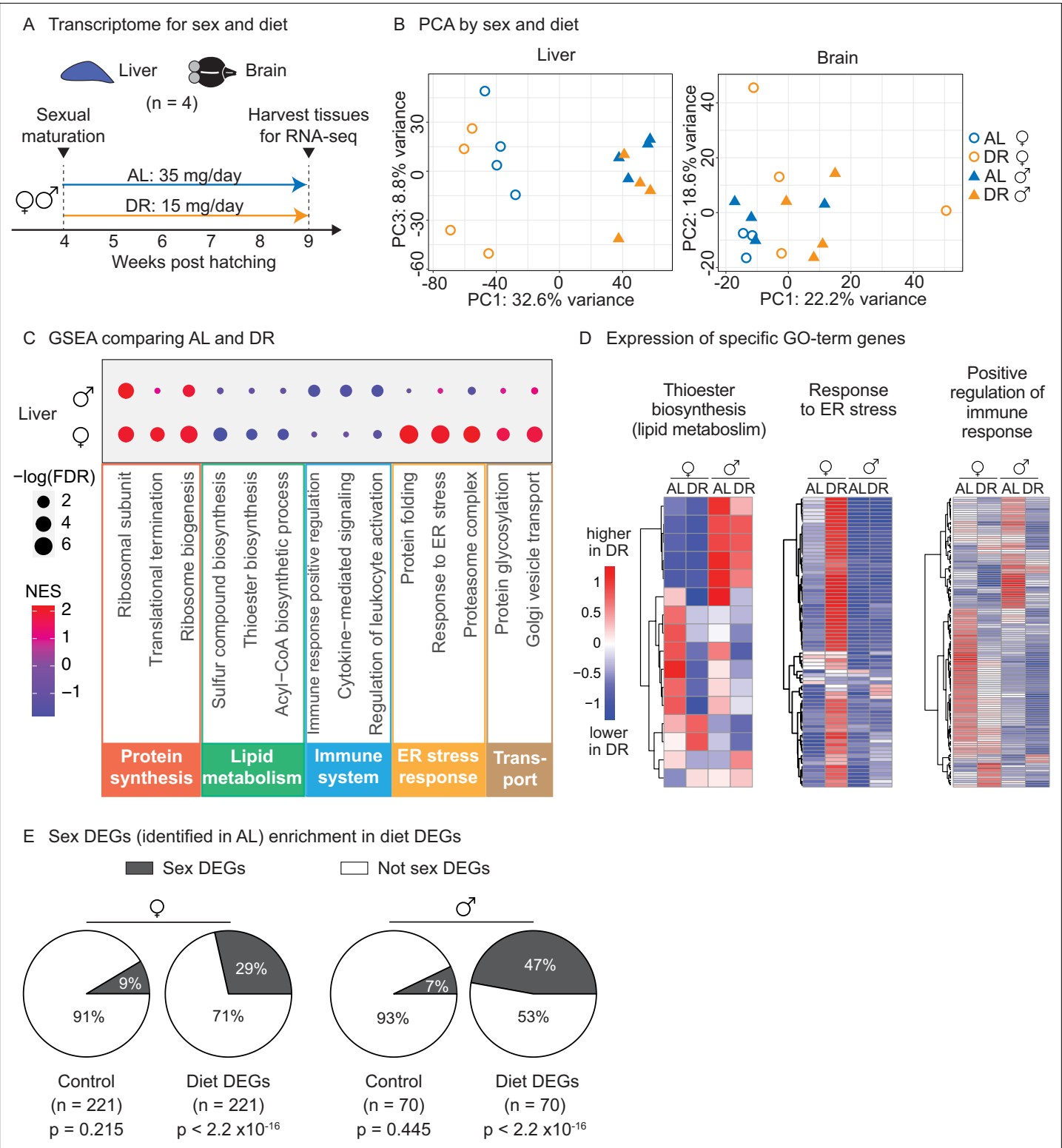

**Figure 5.** Transcriptomic analysis shows that an amount- and time-restricted dietary restriction (DR) induces sex-specific gene expression in the liver of the killifish. (**A**) Experimental scheme to compare the liver and brain transcriptomes in ad libitum (AL) or DR conditions for males and females. A 5-week amount- and time-restricted DR regimen was used (similar to *Figure 4*) and four animals per sex (n = 4) were used. (**B**) Principal component analysis (PCA) of the transcriptomes of all liver samples (PC1 and PC3), or brain samples (PC1 and PC2), with samples coded by sex and diet regimens. For other PCs, see *Figure 5—figure supplement 1A and B*. Variance stabilization transformation was applied to the raw counts, and the transformed count data were used as PCA input. Each symbol represents the full transcriptome of one individual fish. (**C**) Selected gene set enrichment analysis (GSEA)

*Figure 5 continued on next page*

*Figure 5 continued*

results, comparing the AL and DR liver transcriptomes for males and females. Dot color represents the normalized enrichment score of each GO term. Red, higher expression in DR; blue, lower expression in DR. Dot size represents the –log10 of the adjusted p-value (i.e., false discovery rate [FDR] after multiple hypotheses testing). (**D**) Average scaled expression of the genes underlying specific GO terms for the liver. The average scaled expression of the four replicates per condition (Female AL; Female DR, Male AL; Male DR) was plotted in the heatmaps (see 'Materials and methods'). Red, higher expression in DR; blue, lower in DR. (**E**) Sex differentially expressed genes (DEGs) (identified in AL livers) were enriched in diet DEGs, for both males and females. The non-diet DEGs control gene set ('control') and the diet DEGs shared the same transcript expression distribution and gene group size (see 'Materials and methods'). The percentage of genes being sex DEGs for each gene set ('enrichment') was plotted as pie charts. The number of genes in each gene set (n) is indicated in parentheses. Significance determined by a two-tailed Fisher's exact test. For the control gene sets, the median p-value and enrichment in the bootstrapped results were reported.

The online version of this article includes the following source data and figure supplement(s) for figure 5:

**Source data 1.** Experimental metadata for bulk RNA-sequencing experiments.

**Source data 2.** Diet differentially expressed gene (DEG) list (dietary restriction [DR] vs. ad libitum [AL]) for female liver tissues.

**Source data 3.** Diet differentially expressed gene (DEG) list (dietary restriction [DR] vs. ad libitum [AL]) for male liver tissues.

**Source data 4.** Sex differentially expressed gene (DEG) list (male vs. female) for all ad libitum (AL) liver tissues.

**Source data 5.** Sex differentially expressed gene (DEG) list (male vs. female) for all dietary restriction (DR) liver tissues.

**Source data 6.** Gene set enrichment analysis (GSEA) results for female diet differentially expressed genes (DEGs).

**Source data 7.** Gene set enrichment analysis (GSEA) results for male diet differentially expressed genes (DEGs).

**Source data 8.** Gene Ontology enrichment results for diet differentially expressed genes (DEGs).

**Source data 9.** Gene Ontology enrichment results for sex differentially expressed genes (DEGs).

**Source data 10.** Diet–sex interaction differentially expressed gene (DEG) list.

**Source data 11.** Bootstrap data for the control sex differentially expressed genes (DEGs) enrichment in diet DEGs.

**Figure supplement 1.** Transcriptomic analysis shows sex-specific gene expression in response to dietary restriction (DR) in killifish livers.

set size (*Figure 5E*, *Figure 5—figure supplement 1E*). These results indicate that DR preferentially modulates sexually dimorphic genes in the liver, perhaps because of the different metabolic needs of males and females for reproduction. Collectively, these observations indicate that an amount- and time-restricted DR regimen impacts the expression of metabolic and stress/inflammatory pathways in the liver in a sex-specific manner, and this may underlie the beneficial effects of DR on male, but not female lifespan.

## Using automated feeders to develop a positive associative learning assay for the killifish

Food not only influences growth, fertility, and lifespan, but it is also a potent reward across species, including humans (*Lutter and Nestler, 2009*). The rewarding aspect of food has been used for developing feeding-associated learning behaviors in many model organisms, including worms (*Cho et al., 2016*; *Kauffman et al., 2010*; *Lim et al., 2018*; *Stein and Murphy, 2014*), flies (*Das et al., 2014*), zebrafish (*Doyle et al., 2017*; *Pylatiuk et al., 2019*; *Sison and Gerlai, 2010*), mice (*Steinberg et al., 2020*), and non-human primates (*Rolls, 2006*). Assaying learning behavior over an animal's lifetime provides a way of examining functional decline with age. An aversive learning assay has been previously established in killifish and revealed age-dependent decline (*Valenzano et al., 2006a*; *Valenzano et al., 2006b*). However, repetitive exposure to aversive stimuli may induce stress responses that could affect lifespan. It would be helpful to develop a learning behavior assay for killifish that would require little manipulation of an animal and would use an appetitive stimulus rather than an aversive one to limit stress-related responses.

We used our programmable feeding platform to establish a positive associative behavior ('learning') assay, also known as classical conditioning or Pavlovian conditioning. For this assay, we used a red light above the tank as the 'conditioned stimulus' and food delivered by the automated feeder as the 'unconditioned stimulus.' We integrated a red LED light into the automated feeding system and added an individual camera facing the front of each killifish's 2.8 L tank for video recording (*Figure 6A*). The red LED light turns on 2 s after video recording starts, and 7 s later, the food is automatically dropped by the feeder to the water surface (*Figure 6A*).

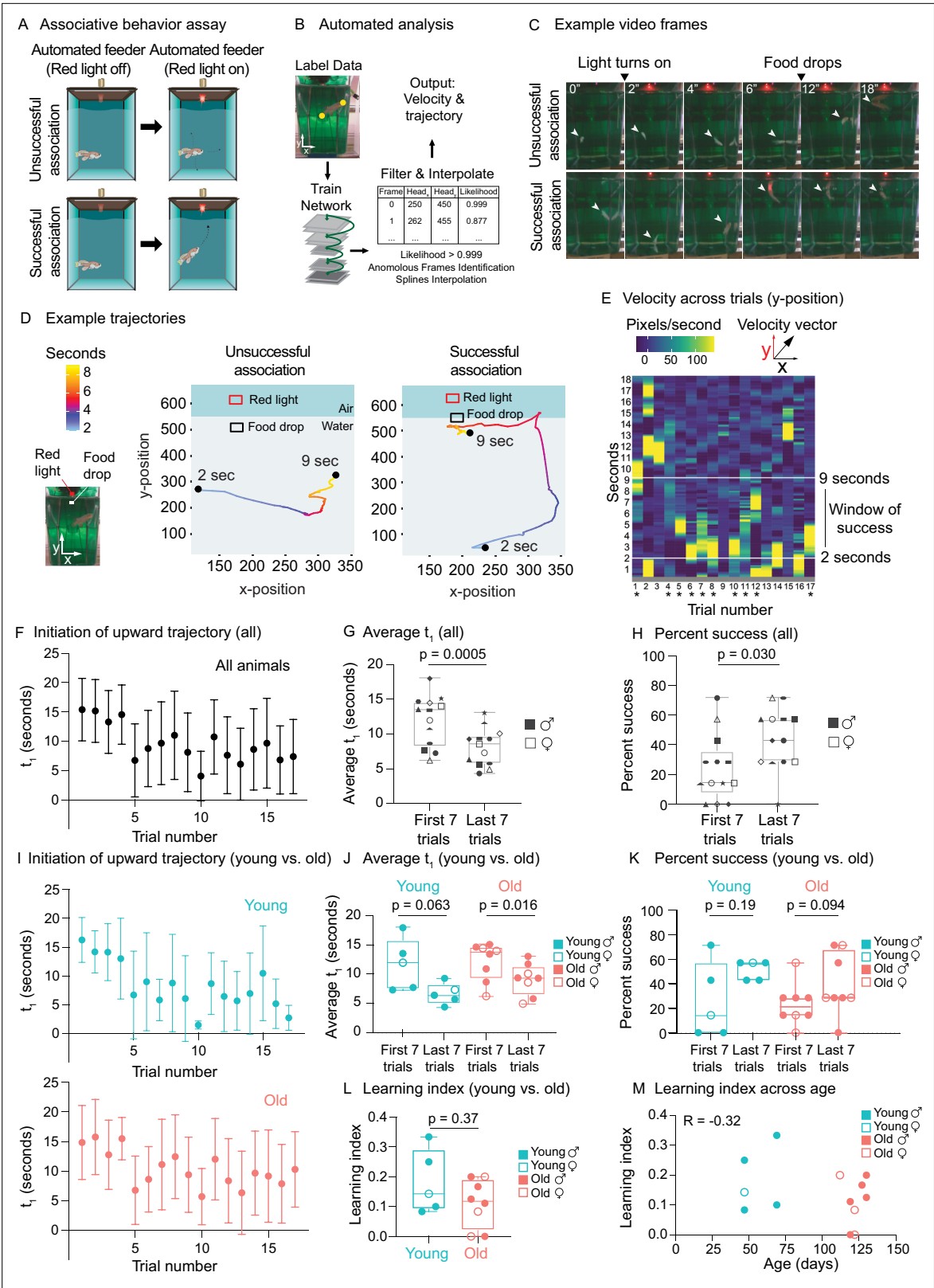

**Figure 6.** Use of automated feeder for assessing positive associative learning behavior. (**A**) Schematic for the killifish positive associative behavior assay. The red light switches on at 2 s after the video recording starts, and the food is dropped at the water surface 7 s later (so 9 s after the recording starts). Unsuccessful association occurs when the killifish does not initiate a surface-bound trajectory within the 7 s window, between the light turning on and the food dropping (so between 2 and 9 s after the recording starts) (top row). Successful association occurs when the killifish initiates a surface-bound

*Figure 6 continued on next page*

*Figure 6 continued*

trajectory within this 7 s window (bottom row). (**B**) Schematic for the automated analysis pipeline. The head and tail positions of the fish and the location of where food was dropped were annotated for selected frames and used to train the DeepLabCut network. After processing all the videos on the trained settings, the results were filtered based on likelihood, anomalous frames were identified and removed, and splines interpolation was performed (see 'Materials and methods'). The resulting head positions of the fish were used for velocity and trajectory calculations. (**C**) Recording video frames from the front of the 2.8 L tank. Top row, an example trial (trial 4) of fish 3, where this fish was unsuccessful in associating food with the red light. Bottom row, an example trial (trial 17) where this fish was successful in associating food with light by moving toward the water surface after the red light turns on and before food is dropped. White arrow points to the fish's position. From the front of the tank, a 2.8 L tank is approximately 14 cm tall and 10 cm wide. (**D**) Trajectory traces generated by the automated pipeline for the videos in (**C**). The x-positions (along the tank width) and y-positions (along the tank height) of fish 3 were tracked at each video frame from 2 s after the recording started to 9 s, and colored according to time. The food drop site indicates where the food arrives at the water surface, and the red light, where the LED light is located. Trial 4, a trial of unsuccessful association; trial 17, a trial of successful association. Sec, seconds. (**E**) The vertical component (y-component) of the fish's (fish 3) velocity, graphed as a heatmap across trial number. The rolling average of 20 frames is plotted for each second of each trial. When the fish exhibits a fast trajectory toward the surface, the pixels/s value is higher (more yellow); whereas when the fish exhibits a slow trajectory toward the surface (or away from surface), the pixels/s value is smaller (bluer). Asterisks indicate successful trials. (**F**) The initiation time of the first surface-bound trajectory ($t_1$) across trials for all fish, quantified by the automated pipeline. Each fish went through 17 trials. The mean $t_1$ for all the fish was calculated for each trial (n = 13), then plotted as a function of trial numbers. A trial was considered successful if $t_1$ was between 2 and 9 s (inclusive). Error bars represent 1 standard deviation above and below the mean $t_1$. (**G**) The average $t_1$ of the first seven trials vs. the last seven trials for each fish, quantified by the automated pipeline. Each symbol represents the average $t_1$ for one fish, and the same symbol denotes the same fish. Males, closed symbols; females, open symbols. Significance determined by a two-tailed Wilcoxon matched-pairs signed-rank test, where the average $t_1$ from the first and last seven trials of the same fish is a matched pair. (**H**) The percentage of successful trials for the first seven trials vs. the last seven trials of each fish, quantified by the automated pipeline. The symbols' shape and color, and the statistics are the same as in (**G**). (**I**) The initiation time of the first surface-bound movement ($t_1$) across trials for young vs. old fish, quantified by the automated pipeline and graphed as in (**F**). Young, blue; old, red. (**J**) The average $t_1$ of the first seven trials vs. the last seven trials, for young (blue) or old (red) fish, quantified by the automated pipeline. Males, closed symbols; females, open symbols. Statistics are the same as in (**G**). (**K**) The percentage of successful trials for the first seven trials vs. the last seven trials, for young (blue) or old (red) fish, quantified by the automated pipeline. Males, closed symbols; females, open symbols. Statistics are the same as in (**G**). (**L**) Learning index for young (blue) and old (red) killifish, quantified by the automated pipeline. The learning index is defined as the inverse of the first trial number for an animal to achieve two consecutive successes. Males, closed symbols; females, open symbols. Significance determined using a two-tailed Wilcoxon rank-sum test. (**M**) Learning index plotted as a function of the fish's age, quantified by the automated pipeline. Males, closed symbols; females, open symbols. R, Pearson correlation coefficient.

The online version of this article includes the following video, source data, and figure supplement(s) for figure 6:

**Source data 1.** Experimental metadata for behavior experiments.

**Source data 2.** Unfiltered raw trajectory data output from DeepLabCut.

**Source data 3.** Filtered and interpolated trajectory data with velocity calculations.

**Source data 4.** Initiation of upward trajectory value ($t_1$) summary for automatic quantification.

**Source data 5.** Fish angular coordinate data for compass plots.

**Source data 6.** Initiation of upward trajectory value ($t_1$) summary for manual quantification.

**Source data 7.** Manual quantification of behavior for female killifish.

**Source data 8.** Manual quantification of behavior for male killifish.

**Figure supplement 1.** Use of automated feeder for assessing positive associative learning behavior.

**Figure 6—video 1.** Example of a trial with a 'successful' association (trial 17, fish 3).
https://elifesciences.org/articles/69008/figures#fig6video1

**Figure 6—video 2.** Example of a trial with an 'unsuccessful' association (trial 4, fish 3).
https://elifesciences.org/articles/69008/figures#fig6video2

We analyzed the video recordings of killifish (a total of 16 males and females, 47–130 days of age) using the automated tracking software DeepLabCut (*Mathis et al., 2018*; *Figure 6B*). We tracked individual fish trajectories (*Figure 6C and D*). An association was defined as 'successful' when a fish initiated a surface-bound trajectory after the red light turned on (2 s after the video recording started) but before the food dropped to the water surface (7 s after the light turned on, so 9 s after the recording started) (*Figure 6C*, bottom, *Figure 6—video 1*, see 'Materials and methods'). An association was defined as 'unsuccessful' when a fish did not initiate a surface-bound trajectory within this window of time (*Figure 6C*, top, *Figure 6—video 2*). Qualitative comparisons of the trajectory traces of individual fish (*Figure 6D*) or compass plots to display trajectory directionality of all fish (*Figure 6—figure supplement 1A*) both suggest that individual fish initiates a surface-bound trajectory more prominently in later trials than in earlier ones. To quantify this surface-bound trajectory, we established

an automated pipeline (*Figure 6B*). We first calculated velocity (i.e., pixels traversed per second) for each fish and assessed the 'bursts' of velocity (yellow regions in *Figure 6E*) for each animal in each trial (see 'Materials and methods'). We then used this metric to plot, for each fish, the time of the first trajectory toward the water surface ($t_1$) to reflect 'initiation of surface-bound trajectory' (*Figure 6F*). For stringent quantification, we only scored the videos of fish that ate during the assay (13 fish out of 16 total) because a successful eating behavior indicates that the fish could see the food at the surface and was motivated enough to move (see 'Materials and methods'). The average time of the first surface-bound trajectory for these 13 fish was significantly shorter in the last seven trials than in the first seven trials (p=0.0005) (*Figure 6G*). The percentage of successful trials in the last seven trials was also significantly higher than that in the first seven trials (p=0.030) (*Figure 6H*). These results were confirmed by manual quantification (*Figure 6—figure supplement 1B–E*), with a correlation of 0.80 for $t_1$ (*Figure 6—figure supplement 1F*). Thus, after several trials, killifish move toward the food delivery after the light turns on but before the food drops, consistent with a positive association between red light and food.

We next compared the association performance between young and old animals (grouping both sexes). We verified that the average velocity of young and old fish did not exhibit overt differences, suggesting that old fish could overall move (*Figure 6—figure supplement 1G*). Old animals (n = 8 males and females, 119–130 days of age) tended to have slightly poorer association performance compared to young ones (n = 5 males and females, 47–70 days of age) (*Figure 6I–K*). Old animals also tended to have a lower 'learning index' – defined as the inverse of the first trial number needed for an animal to achieve two consecutive successes (see 'Materials and methods') (*Figure 6L*). However, at this sample size, the differences between ages were not statistically significant whether animals were binned into young and old categories (*Figure 6J–L*) or not (regression, *Figure 6M*). To detect the observed effect size between young and old (Cohen's d = –0.786) at a p-value of 0.05 and power of 80%, we estimated that 28 animals per age group would be needed when using learning index (and 27 and 132 animals per age group when using average time of first surface-bound trajectory and percentage of success, respectively). Collectively, these results indicate that the automated feeders can be used to determine associative conditioning, a measure of cognitive fitness that could be adapted for future aging studies.

## Discussion

The automated feeding system presented here provides a valuable resource to the killifish community to consistently and reproducibly control the critical factor of feeding. Our system also offers an increased resolution on diet's impact on lifespan in a vertebrate model of aging. The automated feeders improve the precision of food dropped compared to manual feeding, which could reduce variability in experiments (e.g., lifespan studies) and deliver a well-defined diet. Compared to feeding approaches that risk introducing infection (live food) or are labor intensive (manual feeding), our feeding system is a safe and modular solution to feeding not only killifish and other teleost model organisms such as medaka, stickleback, or zebrafish. Furthermore, the tunable frequency and quantity of feedings that can be programmed opens up an experimental space (e.g., night feeding) previously infeasible with conventional feeding approaches. Our design also allows each tank to have its own food hopper and allows straightforward nutrient and drug testing, such as high-fat diets, ketogenic diets (*Newman et al., 2017*; *Roberts et al., 2017*), or dosing of animals with drug-encapsulated food (*Valenzano et al., 2006b*). While there are still areas for optimization, including improved battery life and a simplified user interface, the performance of our automated feeding system is beyond that of manual feeding or several other automated feeding systems (*Doyle et al., 2017*; *Manabe et al., 2013*; *Yang et al., 2019*) in terms of combining precise amount of food dropped, modularity, and scalability. Compared to other recently developed scalable designs (*Lange et al., 2021*), our automated feeders can be added to any tank, including tanks from commercial water systems, allowing flexibility in experimental design.

Using the automated feeding system, we explore dietary parameters and identify a feeding regimen close to an ad libitum diet, as well as a daily dietary restriction diet for the African killifish. We note that the median and maximum lifespan of the killifish in ad libitum conditions in the experiments presented in this study are shorter than some previous studies (*Hu et al., 2020*), though longer than others (*Terzibasi et al., 2008*; *Valenzano et al., 2006b*). These differences might be linked to

food amount or supply differences or other husbandry differences, and 'ad libitum' conditions may in fact vary depending on animal facilities. We also show that the amount- and time-restricted diet we used (~57% dietary restriction) extends the lifespan of male, but not female killifish. Gompertz analysis confirms that dietary restriction reduces the rate of aging in males only. This sex difference in lifespan extension is likely independent of killifish social dynamics, as all animals were housed individually. Lifespan differences between sexes in response to DR are also observed in other species (*Bronikowski et al., 2022*). In mice, extreme DR (40%) extends lifespan in males, but not in females (*Mitchell et al., 2016*), though milder DR leads to a greater lifespan extension in females than males (*Bonkowski et al., 2006*; *Kane et al., 2018*; *Mitchell et al., 2016*). Different strains of mice exhibit more lifespan extension in response to DR in males than in females (*Liao et al., 2010*). In flies, DR has also been shown to impact lifespan differently between the sexes, with female flies enjoying the maximum lifespan benefits from DR at higher calorie levels than male flies (*Magwere et al., 2004*). In *Caenorhabditis elegans*, DR is more beneficial in hermaphrodites than in males (*Honjoh et al., 2017*). Collectively, these observations suggest that sexes respond differently to various DR regimens and that females may not tolerate extreme DR regimens.

We also examine the gene expression changes associated with the sex-specific response to DR in the African turquoise killifish. We find that amount- and time-restricted DR downregulates fatty acid synthesis genes in the liver in females, more so than in males. In mice, DR also downregulates lipid metabolic genes (e.g., *ACYL*, *AACS*) in females (*Hahn et al., 2017*), which has been proposed to protect older animals from visceral fat accumulation (*Kuhla et al., 2014*) and hepatic insulin resistance (*Hahn et al., 2017*). It is possible that in female killifish, the decrease in lipid enzyme expression in the liver in response to the amount- and time-restricted DR regimen is too pronounced and not compatible with sustained metabolic function. DR also upregulates protein folding and ER stress response pathways in female but not male killifish. This was not observed in female mice upon DR (*Hahn et al., 2017*), and this may reflect a response to extreme DR to meet higher secretory demand or protein misfolding stress. Finally, DR downregulates inflammation-related pathways in female killifish, but less prominently than in males. DR was also able to reduce inflammation in female mice (*Swindell, 2008*; *Swindell, 2009*), and this decreased inflammation may contribute to delay the aging process in mammals (*Hunt et al., 2019*; *Maeso-Díaz et al., 2018*; *Swindell, 2009*). But in killifish, the blunted ability of DR to reduce inflammation in females compared to males may also contribute to the lack of lifespan extension by this DR regimen in females. It is possible that the DR regimen defined here is too severe to provide lifespan benefits to females, and AL and DR regimens would likely need to be optimized differently in males and females.

Using the killifish feeding system, we establish a proof-of-concept, positive association learning assay (classical conditioning) that couples food with a red LED light. Combining video cameras with feeders allows many animals to be conditioned simultaneously, at any time, with the ability to collect a wealth of information about animal behavior without investigator-induced disturbance. A positive association learning assay is advantageous over a negative one because it is less stressful for the animal and could be done in conjunction with a lifespan assay without inducing additional perturbation. However, food amounts dispensed for the assay itself would need to be carefully accounted for in the total amount, to avoid inadvertently affecting the overall diet and the lifespan of the animal.

The current version of the positive association assay has limitations. First, it only uses one camera for recording fish behavior, and three-dimensional (3D) behaviors are not fully captured. This assay would therefore be enhanced by using other deep learning methods that better predict 3D poses from monocular (2D) video footage (*Dunn et al., 2021*; *Gosztolai et al., 2021*; *Karashchuk et al., 2021*) or by incorporating multiple cameras calibrated to enable 3D-pose triangulation from multiple perspectives (*Pereira et al., 2020*). Second, while we used a common design for our positive association assay (*Colwill, 2019*), other designs such as the 't$_1$-t$_2$' paradigm, in which animals are first exposed to a stimulus (t$_1$) and learning is then assessed in a testing phase (t$_2$) could improve the assay (*Colwill, 2019*). Such a paradigm can guard against behavioral changes due to fluctuations unrelated to the stimulus, including changes in water parameters, animals' motivation, or circadian activities. Lastly, we verified that older animals were able to eat food (indicating that they could generally see and move) and that there was no significant difference in average velocity between young and old animals. However, we cannot exclude that more nuanced issues with sight or movement ability may contribute to the reduced performance of older animals. Future studies would need to control for these variables

and increase the number of animals when examining the effect of age on learning. Overall, our automated feeding system opens the door to use the killifish as a scalable model to screen for genes or compounds that counter aging and age-related cognitive decline.

## Materials and methods

### Automated fish feeding system

We designed and built an automated fish feeding system that comprises individual battery-powered wireless automated feeders ('feeders'), a co-located server ('local server'), and a cloud-based server ('cloud server').

We designed and built the feeders so that they can be placed on individual 2.8 L tanks supplied by Aquaneering, Inc (San Diego, CA), which are commonly used for killifish and zebrafish husbandry, and fit into the tank lid's two foremost holes. The feeders can precisely deliver a fixed amount of dry fish food (Otohime fish diet, Reed Mariculture, Otohime C1) repeatedly each day on a highly flexible schedule. They are composed of 3D printed parts (Hatchbox PLA, 1.75 mm), transparent 1/16-inch-thick laser-cut acrylic (Amazon.com), nylon screws and nuts (Amazon.com), a 28BYJ-48 stepper motor (Amazon.com), a Wemos D1 mini ESP8266-based development board (Amazon.com), and a custom-printed circuit board (PCB) of a design ordered through OSH Park (Portland, OR). The design incorporates a green LED and corresponding photoresistor that measures food dropped and allows for automatic calibration. The 3D-printed and laser-cut components were designed in Autocad (Autodesk), and PCBs were designed using Eagle software (Autodesk). We assembled feeders in house and programmed the ESP8266 microcontrollers using MicroPython language.

Feeders running MicroPython use the Message Queuing Telemetry Transport (MQTT) communication protocol to connect to the local server, which acts as the MQTT broker, and obtain the current time and feeding regime. Each feeder can be programmed with separate and independent feeding regimes (*Figure 1—figure supplement 1B*). For example, for the AL regimen, the feeder was programmed to deliver seven times per day between 7 am and 7 pm (every 2 hr) and to drop 5 mg of food each time. For the DR regimen, the feeder was programmed to deliver three times per day evenly between 7 am and 9 am and to drop 5 mg of food each time. When several feeders were used in parallel (one for each fish), each of the feeders was programmed independently on the server. Feeders remain in deep sleep power-saving mode until their designated feeding, at which point they rotate their acrylic feeding disc using the onboard stepper motor from under the food hopper, in between the green LED and photoresistor array, and over the drop site, before returning to the food hopper. This releases 5 mg of food per feeding while measuring the resistance of the photoresistor (*Figure 1—figure supplement 1A*, steps 3–6), providing a reportable confirmation of food delivery. The photoresistor readings for each individual device are recorded on a feeding log on the server, confirming that food has dropped, or that there has been a missed feeding. Missed feedings can occur for a variety of reasons, including low levels of food in the hopper or the loss of communication of the feeder with the server. The feeding log records feedings associated with the unique ID of a feeder, so the researcher can be alerted to the issue and rectify it.

After feeding, a confirmation is relayed from the feeder to the local server, using the MQTT protocol, and the confirmation is uploaded to the cloud server via an MQTT bridge. The cloud server acts as the orchestrating database, receiving commands from the user via a Julia language-based command line interface that communicates with the underlying MQTT protocol, the SQLite database of feeder orders, the logs of feeding confirmations, and checked-in feeders. This cloud server exists as an Amazon Web Service EC2 instance and relays changes from users back through the MQTT bridge to the local server and back to individual feeders, while running maintenance scripts that provide status updates on all feeders.

An overview of the system and links to the components (also listed in *Figure 1—source data 1*) is available in a GitHub repository (*McKay, 2021*).

### African turquoise killifish husbandry

All experiments were performed using the GRZ strain of the African turquoise killifish species *N. furzeri*. Animals were housed in a 26°C circulating water system kept at a conductivity between 3500 and 4500 µS/cm and a pH between 6 and 7.5, with a daily exchange of 10% with water treated by

reverse osmosis. All animals were kept on a 12 hr day/night cycle and housed within the Stanford Research Animal Facility in accordance with protocols approved by the Stanford Administrative Panel on Laboratory Animal Care (IACUC protocol # 13645).

Unless otherwise noted, animals were raised as follows: pairs of single GRZ males and single GRZ females between 1 month and 3 months of age were placed in a 2.8 L tank and allowed to breed over a 24 hr period in sand trays placed for embryo collection. After 24 hr, the trays were collected, and embryos were separated from the sand by sieving. Collected embryos were placed in Ringer's solution (Sigma-Aldrich, 96724) with 0.01% methylene blue at 26°C in 60 mm × 15 mm Petri dishes (E and K Scientific, EK-36161) at a density between 10 and 50 embryos per plate. After 2 weeks of monitoring, embryos were transferred to moist autoclaved coconut fiber (Zoo Med Eco Earth Loose Coconut Fiber) lightly packed in Petri dishes (E and K Scientific, EK-36161) at the same density as per the previous 2 weeks and then incubated for another 2 weeks at 26°C. After 2 weeks on moist coconut fiber, hatching was induced by placing embryos in chilled (4°C) 1 g/L humic acid solution (Sigma-Aldrich, 53680) and incubating them in that solution overnight at room temperature. The fry were transferred to 0.8 L tanks at two fry per tank for the first 2 weeks and then one fry per tank for the final 2 weeks on the circulating system (26°C). Fry were fed freshly hatched brine shrimp (Brine Shrimp Direct, BSEP6LB) twice per day for the duration of the first 4 weeks post-hatching. After 4 weeks, adult animals that had inflated swim bladders (identified by the ability to float) were individually housed in 2.8 L tanks where they were fed Otohime fish diet (Reed Mariculture, Otohime C1). Animals were sexed at 4 weeks by visual inspection: males exhibit vivid tail fin colors whereas females do not. Animals were placed on the automated feeding system starting at 4 weeks of age, with the exception of feeder-naïve animals used for the behavioral assays. The animals on feeders were fed for the entirety of the experiments using different regimens including AL (7 feedings of 5 mg per feeding spread throughout the day), DR (3 feedings of 5 mg per feeding in the morning), and overfeeding (12 feedings of 5 mg per feeding spread throughout the day). For behavior experiments, feeder-naïve animals were fed twice a day 20 mg between 8 am and 11 am in the morning and between 2 pm and 5 pm in the afternoon during the week and once 20 mg per day between 8 am and 5 pm during weekends until they were used for the behavioral experiments.

## Comparison between automated and manual feeding

To compare automated feeding with manual feeding, automated feeders were programmed to drop a total of 40 mg Otohime fish diet (eight rotations of the feeding mechanism with 5 mg per rotation). In parallel, individual lab members were instructed to measure out a total of 40 mg of Otohime fish diet (two feeding spoons with 20 mg per spoon). The mass of each manual feeding was measured using a scale to 1 mg precision. The measurement was repeated 9 times per individual and a total of 19 times across four different automated feeders. Estimates of the standard deviation both for automated and manual feedings were calculated by bootstrap using the Bootstrap.jl package in Julia and then converted to precision by taking the inverse of the variance.

## Growth rate experiments and analysis

All fish enrolled in a lifespan experiment were measured for size at two time points: (1) young adult (28 days post-hatching, which is the time of enrollment on the automatic feeders), and (2) death. To avoid concerns that nonlinear growth rates with older aged animals would confound the analysis, only animals that lived 4 months or fewer were considered. To measure the fish size, fish were placed in a clear plastic crossing tank (Aquaneering, ZHCT100T) with a water depth of 3 cm and a reference ruler underneath. Images were taken using a fixed-position digital camera. Analysis of size was carried out by measuring the length of the animals in pixels and converting to length based upon the reference ruler's size in pixels. After animals had died (the day of death or the next day), dead animals were placed on a ruler and their length measured directly using a digital camera. The difference in length divided by the time between the first measurement and death was then reported as the fish growth rate. To accurately compare animal growth rates despite different times of death, only animals that died before 4 months were compared.

Growth rates were compared between conditions using the two-sided Wilcoxon rank-sum test with a significance threshold alpha of 0.05.

## Fertility experiments and analysis

Fish from the same hatching date at 4 weeks of age (AL and DR comparison) or 8 weeks of age (AL and overfeeding comparison) were paired in 2.8 L tanks, with one male and one female. Each animal was given the same feeding regime prior to enrollment: brine shrimp as described above for the first 28 days post-hatching then manual feeding of Otohime C1 fish diet up until the beginning of the enrollment on automated feeders. At the beginning of the enrollment, pairs of males and females were crossed in the afternoon with sand trays and uncrossed ~24 hr later. Pairs were crossed and uncrossed weekly on the same day and crossed until one of the pairs died. Sand trays with embryos were removed at that time and sieved as described above to isolate embryos, which were then rinsed and counted. The total number of embryos produced and the number of fertilized embryos (indicated by separation of the chorion from the yolk membrane) were recorded and reported for each pair for that week.

Fertility was analyzed between feeding conditions by treating each pairing as an independent data point and using two-sided Wilcoxon rank-sum test with a significance threshold alpha of 0.05. Animals used for fertility assessment were part of an independent cohort from that used for the lifespan and growth rate evaluation.

## Lifespan experiments and analysis, including Gompertz curve estimation

Embryos for lifespan experiments were produced as described above. Briefly, GRZ males and females were bred overnight, with embryos placed in Ringer's solution for 2 weeks at 26°C and on coconut fiber for another 2 weeks at 26°C before being hatched in humic acid. Any animals that did not hatch were removed from the experiment. Fry that hatched were transferred to 0.8 L tanks and split on the strict timescale previously described: two fry per 0.8 L tank for the first 2 weeks post-hatching, and then one fry per 0.8 L tank for the subsequent 2 weeks post-hatching. After 4 weeks post-hatching, adult individuals were individually placed in 2.8 L tanks for the remainder of their lives. When placed in 2.8 L tanks, animals were transferred to feeding with Otohime C1 fish diet using an automated feeder. Both males and females were used for lifespan experiments unless otherwise stated, and sex was determined by the bright coloring of the male tail fins. Animals were checked daily and animals that died were recorded as dying on the day found. Two cohorts in total were aged, with animals from each cohort being from the same breeding pairs and enrolled into the cohort based upon hatch date. Any animals whose automated feeders stopped performing due to damage to the feeder were censored (noted as '0' in the 'Observed' column in *Figure 4—source data 1* and *Figure 4—source data 2*).

Statistical analysis of lifespan experiments was conducted using the log-rank test with a significance threshold alpha of 0.05. We also analyzed lifespan data with the Cox proportional hazard model controlling for hatch date (with a significance threshold of 0.05). Gompertz curves were calculated using the combined males from both cohorts 1 and 2. Briefly, males on dietary restriction or ad libitum feeding regimes were pooled and a Gompertz distribution fitted to the two groups using the R flexsurv library and the flexsurvreg function with a 'Gompertz' distribution. The resulting estimated parameters for the intercept and slope of the Gompertz curve were then plotted in Julia using a custom script. Estimates for the hazard rate were assessed by calculating the hazard function based upon 3-week binning of mortality rates for the two groups, DR males or AL males. Gompertz curve parameter estimates are reported with estimated 95% confidence levels.

## Fish cohort and tissue dissection for RNA-seq experiment

All fish were raised from embryos collected from 15 male–female pairs, when these breeders were 9–10 weeks of age (within their reproductive peak). Collected embryos were treated with mild iodine (0.2%, diluted from Povidone-iodine solution [10% w/v, 1% w/v available iodine, RICCA #3955–16]) on day 1 to reduce contamination, incubated in embryo solution with 0.01% methylene blue (Kordo, #37344), and then placed on moist coco fiber on the 14th day after collection. After 2 weeks on coco fiber, fish were hatched in ~1-cm-deep cold 1 g/L humic acid solution, at room temperature. For the next 4 days, the fish were housed on the countertop of the animal facility at 25°C. System water was added to the hatching containers, and fish were fed live brine shrimp daily. For the following 2 weeks, fish were housed at a density of four fish per 0.8 L tank and fed with brine shrimp daily. In the following 2 weeks, housing density was reduced to two fish per 0.8 L tank and fish were fed as before. Each

fish was first imaged from the top using a digital camera at a fixed position and then transferred to a 2.8 L tank. Lastly, an automatic feeder was put on each tank, and either AL or DR feeding regime was assigned randomly to each fish. Feeders were programmed according to this random assignment. Metadata for the animals used in this experiment can be found in *Figure 5—source data 1*.

Brains and livers were harvested from these killifish in two batches on 2 days (four males and four females on each day), when the fish were 9 weeks of age and had been subjected to the assigned diet regimes for 5 weeks. Fish were randomly assigned to a harvest batch, with each batch having two fish from each condition (AL-Male, DR-Male, AL-Female, DR-Female). On harvest day, the automatic feeders were first removed from the tank. Next, 1.5 hr after the room light turned on, for AL treatment, fish were manually fed 17.5 mg of dry food; and for DR treatment, fish were fed 7.5 mg of dry food. This quantity of food corresponded to half of the total daily food amount and maintained the differential ratio of food intake for the two diet regimens. From 12 pm to 3 pm, eight fish were euthanized on ice, and organs were extracted and snap-frozen in liquid nitrogen. All organs were stored at –80°C until RNA isolation.

## RNA isolation, cDNA library generation, and sequencing

We isolated RNA from liver and brain using QIAGEN RNeasy Mini kit (QIAGEN, #74106) according to the manufacturer's instructions. The processing order of each organ was randomized. The tissues were transferred to the 1.2 mL Collection Microtubes (QIAGEN, #19560) on dry ice in a 4°C cold room to reduce tissue thawing. Next, an autoclaved metal bead (QIAGEN, #69997) and 700 µL of QIAzol (QIAGEN, #79306) were added to each tube. Two rounds of tissue homogenization were performed on the TissueLyserII machine (QIAGEN, #85300) at 25 Hz, 5 min each, at room temperature. The lysate was transferred to new 1.5 mL tubes, 140 µL chloroform (Fisher Scientific, #C298-500) was added, and the tubes were vortexed for 15 s. After incubation at room temperature for 2–3 min, lysates were centrifuged at 12,000 × *g* at 4°C for 15 min. The aqueous phase was mixed with 350 µL ethanol (200 Proof, Gold Shield Distributors, #412804) by inverting the tubes 10 times. The mixture was transferred to the RNeasy column, washed with 350 µL RW1 buffer (provided by the RNeasy Mini kit), and treated with DNase I (following the kit's protocol) at room temperature for 15 min. The column was washed two times with 500 µL RPE buffers, and the RNA was eluted with 50 µL nuclease-free water (Invitrogen, #10977023). RNA quality and concentration were measured using an Agilent 2100 Bioanalyzer and the Agilent Nano Eukaryotic RNA Kit (Agilent, #5067-1511). All bioanalyzer assays were performed by the Stanford Protein and Nucleic Acid Facility.

Next, cDNA synthesis was performed using the Takara SMART-seq v4 PLUS kit (Takara, #634889) according to the manufacturer's instructions. Briefly, 10 ng of RNA from each sample was incubated with 1 µL of 10× Reaction Buffer and 2 µL of CDS primer IIA in a 12.5 µL reaction, at 72°C, for 3 min. Next, each sample was mixed with 7.5 µL RT Master Mix (4 µL 5× Ultra Low First-Strand Buffer, 1 µL SMART-Seq v4 48 µM Oligonucleotide, 0.5 µL 40 U/µL RNAse inhibitor, and 2 µL SMARTScribe II Reverse Transcriptase), incubated at 42°C for 90 min and then 70°C for 10 min. To amplify the cDNA, each sample was mixed with 30 µL of PCR Master Mix (25 µL SeqAmp PCR Buffer, 1 µL PCR Primer II A, 3 µL nuclease-free water, and 1 µL SeqAmp DNA Polymerase) and PCR-amplified for a total of eight cycles. The amplified cDNA (a total of 30–50 µL) was purified using 54-90 µL Beckman Coulter AMPure XP beads (Beckman Coulter, #A63880) following the manufacturer's guideline, including two washes of 200 µL 80% ethanol (200 Proof, Gold Shield Distributors, #412804; diluted in nuclease-free water), and elution in 40 µL of nuclease-free water (Invitrogen, #10977023). The concentration and quality of the amplified cDNA library were measured using the Agilent 2100 Bioanalyzer and Agilent's High Sensitivity DNA Kit (Agilent, #5067-4626) by the Stanford Protein and Nucleic Acid Facility.

The cDNA libraries were prepared using the Illumina NexteraXT DNA library prep kit (Illumina, #FC-131-1096) based on the manufacturer's protocol. Briefly, for tagmentation, 0.5 ng of the cDNA (5 µL total) was mixed with 10 µL TD buffer and then 5 µL ATM buffer from the Nextera kit. Each reaction was incubated at 55°C for 4 min and then cooled to 10°C. To stop the tagmentation reaction, 5 µL of NT buffer was added immediately and incubated with the reaction mixture at room temperature for 5 min. The Illumina Nextera XT Index Kit v2 (Illumina, #131-2001) was used for library indexing. The cDNA library was indexed and amplified for a total of 13 cycles in a PCR reaction that contained the 20 µL of tagmented DNA, 5 µL of i7 adaptor, 5 µL of i5 adaptor, and 15 µL NPM buffer. The amplified cDNA library (50 µL total) was purified using 90 µL of Beckman Coulter AMPure XP beads as described

above and eluted in 52.5 µL RSB buffer. The concentration and quality of the library were measured using the Agilent 2100 Bioanalyzer and Agilent's High Sensitivity DNA Kit (Agilent, #5067-4626) by the Stanford Protein and Nucleic Acid Facility.

The cDNA library was sequenced on an Illumina NovaSeq 6000 (2x150 bp paired-end) by Novogene (Novogene, Beijing, China), at a sequencing depth of >20 million pair-end reads. We noted that some of the samples were sequenced on a separate flow cell, which were indicated in the 'sequencing batch' column of the sample metadata file.

## Sequencing quality control and read mapping

Raw sequencing data (FASTQ files) were processed and checked for quality using Trim-galore v0.4.5. The processed reads were aligned to the African turquoise killifish reference genome downloaded from NCBI (Nfu_20140520, GCF_001465895.1). The read alignment was performed using STAR v2.7.1a (*Dobin et al., 2013*) with the default parameters. Out of 32 RNA-seq samples, 15 samples had >90% of reads mapped to the genome; 10 samples, 80–90% reads mapped; and 7 samples, 75–80% mapped. Next, we used samtools v1.5 (*Li et al., 2009*) to remove the reads that map to multiple genomic regions, using the parameters of MAPQ < 255 ('samtools view -q255 -b'). These uniquely mapped reads were then used to generate the read counts for each gene, for which we used the 'featureCounts' program from subread v2.0.1 (*Liao et al., 2014*) using the default parameters.

## DESeq2 analysis

We used DESeq2 v1.32.0 (*Love et al., 2014*) to analyze the differential gene expression as follows. First, a pseudocount of '1' was added to all the read counts. Next, the 'dds' object was created using the parameter 'design = ~Condition', where 'Condition' is a compound variable specified by 'tissue_sex_diet'. The genes were filtered such that those with a sum across all samples fewer than 16 counts were removed from the 'dds' object.

PCA was performed by first applying the variance stabilizing transformation ('vst') on the 'dds' object and then visualizing using ggplot2 v3.3.5. To generate the PCA plot for a given tissue, the count matrix (already added a pseudocount of '1') was first subsetted so that only the samples for that tissue remained.

## Diet DEGs and sex DEGs for the liver

Because tissue identity was the main contributor of the variation among samples (as expected), we performed the DESeq2 analysis for only the liver samples, which allowed us to identify more subtle gene expression changes that might be explained by sex and/or feeding regimen. For the brain samples, one AL sample was located 3 standard deviations away from the rest of the samples in PC1. We thus excluded this sample from the PCA plots.

To identify the diet differentially expressed genes (diet DEGs) for the liver, we used the tissue-subsetted 'dds' object created above ('design = ~Condition') and extracted the DEGs between the AL (reference) and DR samples, separately for males and females, using the 'contrast' parameter of the 'results' function in DESeq2 (*Figure 5—source data 2* and *Figure 5—source data 3*). We used the two gene lists, separately, as the input for the GSEA below.

To identify the sex differentially expressed genes (sex DEGs) for the liver, we extracted the DEGs between males and females under the AL or DR condition, again using the 'contrast' parameter of the 'results' function in DESeq2 (*Figure 5—source data 4* and *Figure 5—source data 5*). There were 1081 sex DEGs in AL and 2151 sex DEGs in DR (p<0.05), with 743 genes shared by the two conditions.

Lastly, we identified the diet–sex interaction genes by creating a 'dds' object for the liver using 'design = ~diet + sex + diet:sex', removing genes with a sum across all samples fewer than 16 counts and running the 'DEseq' function with the 'female AL' condition as a reference. A positive log2-transformed fold change (log2FC) occurs when the ratio of DR/AL is higher in males than in females; a negative log2FC occurs when the ratio of DR/AL is lower in males than in females. There were 12 significant diet–sex interaction genes (*Figure 5—source data 10*). While this number of genes is too low for GO enrichment analysis, these genes are involved in immune-related, lipid metabolism, and intracellular trafficking function, consistent with the GSEA results.

To visualize the expression distribution (*Figure 5—figure supplement 1C*), we used the 'plotMA' function in DESeq2 to make a scatterplot for the log2FC and the log10 of the mean normalized counts.

## Enrichment of sex DEGs in diet DEGs

We calculated the enrichment of the sex DEGs (identified either in AL or in DR) in diet DEGs as follows. First, we removed the genes with an 'NA' value for the p.adjust in any dataset, and then we created a contingency table (for males and females separately) by counting the genes that satisfied the following categories: (1) group 1: genes differentially expressed by both sex and diet (i.e., sex DEGs with p.adjust<0.05 and diet DEGs with p.adjust<0.05); (2) group 2: genes differentially expressed only by sex (i.e., sex DEGs, p.adjust<0.05) and not by diet (i.e., diet DEGs, p.adjust≥0.05); (3) group 3: genes differentially expressed only by diet (i.e., diet DEGs, p.adjust<0.05) and not by sex (i.e., sex DEGs, p.adjust≥0.05); and (4) group 4: genes not differentially expressed by diet or sex (i.e., diet DEGs with p.adjust≥0.05 and sex DEGs with p.adjust≥0.05). In *Figure 5E* and *Figure 5—figure supplement 1E*, the enrichment was defined as the ratio of group 1 gene number over the number of diet DEGs (i.e., (group 1)/(group 1 + group 3)). Because the two sexes were analyzed separately in the DESeq2 analysis, two contingency tables were created: one for male diet DEGs and one for female diet DEGs. A two-tailed Fisher's exact test was used to calculate the statistical significance of the enrichment.

Non-diet DEG control gene sets (one for either sex) were generated to control for any bias in gene expression distribution and gene group size of the diet DEGs. To do so, for each diet DEG, we identified all the non-diet DEGs that were within 2% of the diet DEG's average expression level (average across all samples). When the expression threshold was set at 1%, one diet DEG had no associated control gene; thus, an expression threshold of 2% was selected. These genes constituted the 'non-diet DEG control group' for a given diet DEG. Next, we randomly selected one gene from each non-diet DEG control group, for all 221 of the female diet DEGs and 70 for the male diet DEGs. These randomly selected genes comprised the control gene set, which had the same number of genes as the diet DEGs. To perform a boostrapped analysis, we generated 1000 control gene sets and tested each of them for sex DEG enrichment by a two-tailed Fisher's exact test. We plotted the median enrichment and the median p-value (Fisher's exact test) of these 1000 iterations. All pie charts were plotted in R v4.1.0. The bootstrap data can be found in *Figure 5—source data 11*.

## Gene set enrichment analysis (GSEA)

We performed GSEA (*Mootha et al., 2003*; *Subramanian et al., 2005*) as follows. First, for a given gene list (male diet DEGs and female diet DEGs), we sorted all transcripts in descending order based on their ranked scores, which were calculated by multiplying the -log10(p-value) with log2FC. Genes without p-values reported from DESeq2 analysis were removed. Next, we identified the human ortholog name for each killifish gene via protein blast (best hit protein with BLASTp E-value>1e-3). If multiple killifish paralogs were blasted to the same human gene, the average of the ranked scores was calculated and assigned to this human gene name. If no human ortholog was found, this killifish gene was removed from the gene list. Lastly, we ran the enrichment analysis via clusterProfiler v4.0.5 (*Yu et al., 2012*) and the Bioconductor annotation data package (org.Hs.eg.db v3.13.0) and GOstats package v2.58.0. The p-values of the enriched pathways were corrected for multiple hypotheses testing using the Benjamini–Hochberg method (p.adjust). A GO term was considered significantly enriched if it had a p.adjust value <0.05. The top GO terms from the 'male diet DEGs' and 'female diet DEGs' gene lists were graphed as a dot plot. The full GSEA data are given in *Figure 5—source data 6* and *Figure 5—source data 7*.

## Hypergeometric GO enrichment analysis

We used the GSEABase v1.54.0 and GOstats v2.58.0 packages for this analysis. For male diet DEGs and female diet DEGs, we first separated these two diet DEG lists (p.adjust<0.05) based on upregulation or downregulation in DR. The resulting four gene lists were individually used for the hypergeometric test implemented in GOstats v2.58.0. For the sex DEGs identified in AL and those identified in DR, we separated these two sex DEG lists based on upregulation or downregulation in males. Each of the four gene lists was used for the hypergeometric test. The background genes ('universe') were

defined as all of the genes with a non-NA value for p.adjust for a given comparison. The full GO analysis results are given in *Figure 5—source data 8* and *Figure 5—source data 9*.

We found that protein homeostasis (e.g., 'ERAD pathways') and intracellular trafficking (e.g., 'protein glycosylation') GO terms were significantly enriched in the female diet DEGs upregulated in DR (*Figure 5—figure supplement 1D*). In contrast, lipid metabolism (e.g., 'fatty acid biosynthetic process') GO terms were enriched in the female diet DEGs downregulated in DR. For males, intracellular trafficking GO terms (e.g., 'regulation of protein secretion') were also significantly enriched in the diet DEGs upregulated in DR, but the enrichment score was lower than for females. Lipid metabolism GO terms were also enriched in the diet DEGs downregulated in DR in males, but both the enrichment score and the statistical significance were lower than for females. Thus, the specific upregulation of protein homeostasis pathways in female DR livers and the downregulation of lipid metabolism in female DR livers (to a greater extent than in males) are consistent with the GSEA results.

Lastly, GO analysis on the sex DEGs identified in AL showed that the genes more highly expressed in AL male livers were enriched in cellular amino acid catabolic processes and gluconeogenesis pathways. In contrast, genes more highly expressed in AL female livers were enriched in estrogen response, humoral immune response, and female sex differentiation. GO analysis on the sex DEG identified in DR showed that while most GO terms were similar to those for the AL sex DEGs, protein homeostasis pathway (e.g., 'ER stress response') was uniquely enriched in the genes more highly expressed in DR female livers. This observation is consistent with our results from GSEA and GO analysis on the diet DEGs.

## Heatmap

Heatmaps were generated using Pheatmap v1.012. We identified the killifish orthologs for all the human genes associated with each GO term. For each killifish ortholog gene in the GO term, we subjected the normalized counts generated from the DESeq2 analysis to the following. First, we calculated the mean and standard deviation (SD) across all the replicates (n = 4 for AL; n = 4 for DR) of the selected gene. Next, we scaled each normalized count value ($x_i$) by $z_i = (x_i - mean)/SD$ and used $z_i$ as the input for heatmaps. Hierarchical clustering was performed using Pearson correlation and 'ward.D2' method (hclust(as.dist(1-cor(t(allCounts_GeneGO_plot_f))), method = 'ward.D2')). Heatmaps were plotted using the distance matrix (from clustering) and the parameters (clustering_method = 'ward.D2', treeheight_row = 0, treeheight_col = 1, cluster_cols = F, cluster_rows = T).

## Fish cohorts for the associative learning assay

Animals were raised as described above, except that GRZ pairs were crossed for a longer duration (up to 1 week) before embryo collection. The animals were raised to adulthood with less stringent density requirements: 4–5 fry were placed per 0.8 L tank for 2–3 days before being split to two fry per tank for the remainder of the 4 weeks before sexual maturity. After sexual maturity, fish were individually housed in 2.8 L tanks. For these experiments, fish were manually fed as previously described ('feeder-naïve') and consisted of males (n = 17) and females (n = 10) with age ranging from 36 days to 130 days post-hatching. Metadata for the animals used in this experiment can be found in *Figure 6—source data 1*.

## Associative learning assay

Animals were transferred from their home tanks to designated behavioral tanks (on a different rack) in batches, with individuals selected for each batch chosen at random. Individual tanks were separated from one another using white foam-core sheets. The tanks on the behavior rack received recirculating water from the system as did the home tanks.

Each experiment consisted of 17 trials conducted across three consecutive days, with 7 trials for the first 2 days and 3 trials on day 3. On day 0 (the day before the first trial), animals were placed in individual behavior tanks in the afternoon after receiving their afternoon manual feeding. Each automated feeder used for the behavioral assay had an additional red LED light that served as the conditioned stimulus and was installed adjacent to the feeding hole. This red LED light is programmed to turn on and off by the same Julia language-based command line interface that communicates with MQTT protocol that sends feeding commands to each feeder. We programmed the automatic feeders to initiate the first session of stimulus presentation and feeding on day 1 at 7 am. For the sessions on

days 1–3, the feeders were programmed to run the sessions autonomously. On day 4, animals were removed from behavior tanks and placed in their home tanks, and normal feeding and care resumed.

Videos were recorded using an ESP32-CAM module from the anterior of the tank. The sequence of events recorded was as follows: (1) camera turned on, (2) on the 2nd second, the red LED light turned on, (3) on the 9th–11th second, the food arrived at the water surface, and (4) camera remained on during feeding and ended >10 s after food dropping. In sum, animals were trained with a conditioned stimulus (red light) delivered 7 s before the unconditioned stimulus (food, 5 mg dropped at a time). Videos were analyzed by manual scoring and automated tracking. Out of the 27 fish tested in this assay, 11 fish were excluded due to technical issues (camera failed to record all 17 trials for 5 fish, and feeders failed to drop food for 6 fish). Three fish displayed eating behavior in fewer than three trials, and they often stayed at the bottom of the tanks and rarely moved. We considered that these fish potentially had underlying health issues, lacked motivation for food, and/or failed to adapt to the videotaping tanks. Thus, we excluded these animals from our analysis. Altogether, a total of 13 animals were analyzed (nine males and four females).

## Automated analysis of the associative behavior assay using DeepLabCut

Automated tracking was performed by DeepLabCut 2.2 (*Mathis et al., 2018*), an open-source deep-learning tracking software, and the output data were analyzed using custom R and Julia scripts. Briefly, the DeepLabCut software was trained on representative annotated training video frames (20 frames/s; the given number of videos annotated was set by the default setting in DeepLabCut) using a transfer learning approach as follows: superficial layers of the neural network were updated while the core ResNet-50 layers were left untouched, resulting in relatively rapid training of the network. All killifish training videos were processed using this network. DeepLabCut results were exported as CSV files, providing the x- (side-to-side in tank) and y- (top-to-bottom in tank) coordinates and likelihood values for fish snouts, fish tails, the food drop site, and the red training light. Tracked videos varied slightly in length, but minimally were 18 s long (360 frames).

Several prefiltering steps were performed on these positional values as follows: (1) first, fish head likelihood values were selected if they passed the threshold of 0.999, meaning that the coordinates for the frames with likelihood values <0.999 were set to 'NA' (leading to exclusion of 41,623 frames). (2) Second, of the selected coordinates, the Euclidean distance between the x,y positions from contiguous frames was calculated for all frames and for all videos. If the Euclidean distance between any two contiguous frames was in the top 5% of all distances calculated, this was considered an anomaly. The coordinates of the second frame of the two contiguous frames were set to 'NA' (after this step, 10,051 additional frames were filtered out). (3) Lastly, given that the previous filtering steps led to missing positional values in the data ('NA'), the final processing step was interpolation. We performed interpolation using the na.spline function in the zoo R package (version 1.8–10). The na.spline function performs cubic spline interpolation, which involves finding low-degree polynomial curves that connect data points and results in smooth, continuous values and matching first and second derivatives where the polynomial curves meet. We did not perform any extrapolation, so some 'NA' values did remain where an unknown value was not preceded or followed by a known value, for example, in cases where there were long stretches of 'NA' in the beginning or the end of a video. The resulting data were used for velocity calculations and plotting of movement trajectories. All the unfiltered raw trajectory data can be found in *Figure 6—source data 2*, and the filtered and interpolated trajectory data with kinematics calculations can be found in *Figure 6—source data 3*.

## Automated scoring and trajectory analysis

The prefiltered and interpolated coordinates were used to calculate the instantaneous velocity in the y-direction (up-down in the tank) for each frame. Rolling averages of the instantaneous velocities were calculated over 20 frames using the zoo package, and then the raw velocity of each frame was adjusted to the 20-frame rolling average. This adjustment allowed us to better visualize the regions of increased activity. We plotted the velocities as heatmaps. For the velocity heatmap, the top 7% and bottom 10% of velocities are set as the extremes on the heatmap color scale. We also used the coordinates to determine the Euclidean distance of the fish from the coordinates of where food drops

for each frame. For each fish, the location of where food drops on the water surface was determined by taking the average of the highest likelihood food drop coordinates across all videos for that animal.

We calculated $t_1$ by (1) determining the beginning and end times of the 'bursts' of velocity activity (yellow regions in *Figure 6E*) for each animal in each trial and (2) determining the times during each trial when the fish was 'arriving at the surface.' We set the threshold for a high-velocity burst for each animal as approximately the top 25% of the velocity range and the threshold for arriving at the surface was defined as 100–150 pixels away from the food drop location (see *Figure 6—source data 4*). The $t_1$ values were calculated by finding the closest velocity burst to a time of arrival at the surface and using the beginning time of the velocity burst as the $t_1$. If the fish never arrived at the surface during a trial, the $t_1$ would be set to 18 s. If the fish began the trial at the surface, the $t_1$ would be set to 0 s. In three trials (out of 221 trials for all fish), the velocity thresholds were set slightly too high, and the velocity bursts were determined by inspecting the velocity heatmap manually (see *Figure 6—source data 4* for details). Finally, we calculated the Pearson correlation between the automated $t_1$ values and the manual $t_1$ values using Prism v.9.3.1.

Fish x-y trajectories (in pixel space) were plotted using the fish head coordinates and were shown relative to the locations of food delivery (calculated as the average of all the high-likelihood locations) and the red light (calculated as the average of all the high-likelihood locations). The trajectory of the fish was colored by time, with the blue representing the fish location at 2 s and yellow, at 9 s. The trajectories from trials 4 and 17 for fish 3 were plotted. Plots were made using the 'geom_path()' function in R. All the data related to automatic quantification can be accessed in *Figure 6—source data 4*.

## Compass plot

A compass plot displays the angle of a velocity vector, showing the fish's direction of movement between the initial position (when the light turns on at 2 s) and the final position (when the food drops at 9 s). The compass plots shown in *Figure 6—figure supplement 1A* are the frequency distribution of the angular coordinates (see below) for all the velocity vectors that were annotated for all the fish in the first seven trials or annotated for all the fish in the last seven trials. To generate these plots, we first determined the initial position of the fish after the red light turned on (2 s) and the final position of the animal when the food was dropped (9 s). Next, we converted these positions from Cartesian coordinates (x, y) to polar coordinates (r, θ). Lastly, we converted the angular coordinate (θ) of the vector (from the initial position to the final position) from radians to degrees. The angular coordinates (θ) from the first seven trials (or the last seven trials) for all the fish were pooled and then binned to create a frequency distribution (bin size = 20°, histograms centered on the midpoint of each bin) that could then be plotted as a compass graph using the 'coord_polar()' function in R. Each 'wedge' on the compass plot represents a bin. The wedge is directed toward the angle (out of 360°) that corresponds to the midpoint of the bin, and has a radius that indicates the frequency of angles in the bin (larger radius indicating that more velocity vectors fallen into that bin). All the velocity-vector bins have the same origin. All plots were generated using ggplot2 in R. All data for input into the compass plots can be found in *Figure 6—source data 5*.

## Manual analysis of the associative behavior assay

For manual analysis, each video (one trial) was analyzed second-by-second, and three metrics were tracked for each second: (1) Did the midpoint of the fish have a positive y-axis displacement during this second? A score of '1' was given if the y-axis displacement was greater than 0 (i.e., the fish moved toward the water surface); otherwise, a score of '0' was given (i.e., the fish moved away from the water surface or stayed at the same water level). (2) Was the fish at the water surface? A score of '1' was given if the fish's snout was within ~0.5 cm of the water surface (the fish's reflection was visible); otherwise, a score of '0' was given. (3) Was the fish eating the food? If so, 'eating' was noted for this second.

For each fish, we graphed metric 1 and 2 scores against time, respectively (*Figure 6—figure supplement 1B*). We identified the first time at which the fish reached the water surface, and then we backtracked when the fish initiated a continuous trajectory toward the surface (i.e., identifying when the string of '1' in the y-axis displacement plot began). This initiation time of the first surface-bound trajectory is defined as '$t_1$.' Sometimes, the fish would stop moving for ≥1 s during its ascent to the surface. In this case, we disregarded this first part of ascension and reported $t_1$ to be the initiation time of the second part of the ascent, during which the fish had a continuous upward trajectory

and ultimately reached the surface. Our measurement in this scenario is likely conservative, and it is possible that the fish moved toward the water surface earlier than what we reported here.

To measure how robust our method is for determining $t_1$, three researchers (EKC, JC, RCK) independently analyzed the same set of 17 videos from one fish. The results were consistent across all three researchers, with a Pearson correlation of >0.95 for any pair-wise comparison, suggesting that this analysis method is robust against the subjective judgment of observers.

## Quantification of successful association

We defined a successful trial to be when the fish initiated its surface-bound trajectory ($t_1$) before food drop. Thus, a successful trial would be the one with $2 \leq t_1 \leq 9$, where '2' corresponded to when the light turned on at 2 s, and '9' corresponded to when the food arrived on the water surface at 9 s. We noted that there was some variability to when the food arrived (the food was set to dropped at 5 s, but there were network and mechanical delays), ranging from 9 to 11 s. Thus, the window of success (2–9 s) should be considered conservative.

To quantify successful association, we calculated the average $t_1$ for the first seven trials ('early trials') and the last seven trials ('late trials') for each fish. The average $t_1$ values for all the fish, or for young and old fish, were plotted as a dot plot overlaid on a box plot, where each fish was denoted as a unique symbol. In the automated quantification method, for two animals (fish 13 and 14), the $t_1$ of trial 7 was not calculated due to low-likelihood position values, and the average $t_1$ of the early trials was calculated using the first six trials.

In another metric to measure successful association, we calculated the number of successful $t_1$, which fell in the range of 2–9 s (including 2 and 9) for the early and late trials, respectively, for each fish. The percentage of success was calculated as the number of successful $t_1$ divided by 7. The values of the average $t_1$, as well as the percent success, for the early trials and the late trials for the same fish were considered as a matched pair for the two-tailed Wilcoxon matched-paired signed-rank test. All the graphs and statistics were performed in Prism v9.3.1.

## Learning index

We defined the 'learning index' as the inverse of the first trial number needed for an animal to achieve a given number of consecutive successes. This metric treats the trials as dependent on one another to reflect that the prior experiences of an animal can influence future performance. We took the inverse of this first trial number to make 'faster association' be a higher learning index value. As an example, for fish 3 (*Figure 6E*), its learning index corresponded to the first trial when two consecutive successful trials occurred. In this case, trial 4 was the first time when $2 \leq t_1 \leq 9$ for two consecutive trials (both trials 4 and 5 were successful). Thus, the 'learning index' would be 1/4 (0.25). Most of the killifish (85%) can achieve two consecutive successes, while only 54 and 38% can achieve three or four consecutive successes, respectively (*Figure 6—source data 6*, 'learning index_3' and 'learning index_4'). Thus, the number of consecutive successful trials reveals the fish-to-fish variation in the association performance across trials, a useful metric for examining the behavior stability of individual fish.

Prism v.9.3.1 was used to generate all dot plots, scatterplots, and Pearson correlation calculation. All the data related to manual quantification can be accessed in *Figure 6—source data 6*, *Figure 6—source data 7*, and *Figure 6—source data 8*.

## Average velocities for young vs. old fish

To calculate the average velocities for each fish, we used the calculated 20-frame rolling average velocities for each frame (see details above). We set the top and bottom 15% of velocities for a given animal to one of two boundary values – the top 15% percentile velocity or the lowest 15% percentile, respectively. Thus, the values outside of the middle 70% were capped to the boundary values. This thresholding prevents the average values from being skewed by poor velocity estimation from the frames with low likelihood values. We used the average of all the 20-frame rolling average velocities across all the frames and all trials for a given animal.

## Cohen's d effect size calculation

We calculated the standardized mean difference (Cohen's d = (mean x1 – mean x2) / standard deviation), cohen.d() and 95% confidence intervals in R. Using this, the effect size estimate of the learning

index calculated from the automated analysis pipeline is –0.7862264 with a confidence interval of (–2.0860663, 0.5136134), grouping both sexes. This is ~1 standard deviation between the two group means.

## Power analysis

Grouping both sexes, the mean and standard deviation of the young group (47–70 days of age post-hatching) and the old group (119–130 days of age post-hatching) were calculated. Power analysis was performing using G*Power v3.1.9.6 with the following parameters: test family: '$t$-tests'; statistical test: 'Means: Wilcoxon–Mann–Whitney test (two groups)'; type of power analysis: 'A priori: Compute required sample size – given α, power, and effect size'; tail(s): 'Two'; parent distribution: 'Normal'; effect size d: '–0.7862264' for learning index (Cohen's d), and '0.8046403' and '0.3544401' for the difference in $t_1$ and in the percentage of success, respectively, between the first and last seven trials (both effect sizes were calculated in G*Power using the mean and standard deviation of the two age groups); α error prob: '0.05'; Power (1-β err prob): '0.8'; and allocation ratio N2/N1: '1.'

Code is available in a GitHub repository, copy archived at swh:1:rev:451bc5d78cd266a00b53612585d201d404f73920 (*McKay et al., 2022*).

## Acknowledgements

We thank Itamar Harel, Brittany Demmitt, Robin Yeo, Ravi Nath, Adam Reeves, Xiaoai Zhao, and Ariana Sanchez for scientific discussion and feedback on the manuscript. We thank Brandon D Kim for providing the raw videos to make Figure 1—video 1. We thank Robin Yeo and Xiaoai Zhao for help with independent code validation. We also thank Susan Murphy, Ben Machado, Rogelio Barajas, Natalie Schmahl, Jacob Chung, and Jadon Shen for their assistance with killifish husbandry. Supported by RF1AG057334 (AB), R01AG063418 (AB), Stanford Brain Rejuvenation Program (AB, TWC), Stanford Graduate Fellowship (AM), a Helen Hay Whitney Fellowship (CNB), and a Jane Coffin Childs Fellowship (JC).

## Additional information

### Funding

| Funder | Grant reference number | Author |
|---|---|---|
| Stanford Brain Rejuvenation Program | | Tony Wyss-Coray |
| Stanford Graduate Fellowship | | Andrew McKay |
| Helen Hay Whitney Fellowship | | Claire N Bedbrook |
| National Institutes of Health | RF1AG057334 | Anne Brunet |
| National Institutes of Health | R01AG063418 | Anne Brunet |
| Jane Coffin Childs Memorial Fund for Medical Research | 61-1762 | Jingxun Chen |

The funders had no role in study design, data collection and interpretation, or the decision to submit the work for publication.

### Author contributions

Andrew McKay, Conceptualization, Data curation, Software, Formal analysis, Validation, Investigation, Visualization, Methodology, Writing – original draft, Writing – review and editing, created the killifish feeding system and conducted and analyzed the validation; Emma K Costa, Conceptualization, Data curation, Software, Formal analysis, Validation, Investigation, Visualization, Methodology,

Writing – original draft, Writing – review and editing, EKC and JC performed and analyzed RNA-seq experiments. EKC designed the automated quantification pipeline and quantified the conditioning assay; Jingxun Chen, Conceptualization, Data curation, Software, Formal analysis, Validation, Investigation, Visualization, Methodology, Writing – original draft, Writing – review and editing, EKC and JC performed and analyzed RNA-seq experiments. JC designed the final manual quantification pipeline and quantified the conditioning assay; Chi-Kuo Hu, Data curation, Validation, Investigation, Visualization, had intellectual input and provided earlier figure iterations. C-KH helped with independent code validation and provided intellectual input. C-KH commented on the manuscript; Xiaoshan Chen, Data curation, Investigation, assisted in manual scoring early iterations of the killifish conditioning assay. XC commented on the manuscript; Claire N Bedbrook, Data curation, Validation, Visualization, validated killifish feeding system by generating independent feeders. CNB helped with independent code validation and provided intellectual input. CNB commented on the manuscript; Rishad C Khondker, Data curation, Investigation, helped JC with manual quantification of the conditioning assay. RCK commented on the manuscript; Mike Thielvoldt, Methodology, provided consultation on aspects of the electrical design. MT commented on the manuscript; Param Priya Singh, Software, Supervision, Validation, Visualization, Methodology, helped with independent code validation and provided intellectual input. PPS guided the analysis of the RNA-seq dataset. PPS commented on the manuscript; Tony Wyss-Coray, provided intellectual input and provided guidance for EKC. TWC commented on the manuscript; Anne Brunet, Resources, Software, Supervision, Funding acquisition, Validation, Methodology, Writing – original draft, Writing – review and editing

### Author ORCIDs
Andrew McKay http://orcid.org/0000-0002-9179-5018
Emma K Costa http://orcid.org/0000-0002-9431-6852
Jingxun Chen http://orcid.org/0000-0001-7320-8652
Tony Wyss-Coray http://orcid.org/0000-0001-5893-0831
Anne Brunet http://orcid.org/0000-0002-4608-6845

### Ethics
All animals were housed within the Stanford Research Animal Facility and treated in accordance with protocols approved by the Stanford Administrative Panel on Laboratory Animal Care (protocol # APLAC- 13645).

### Decision letter and Author response
Decision letter https://doi.org/10.7554/eLife.69008.sa1
Author response https://doi.org/10.7554/eLife.69008.sa2

## Additional files

### Supplementary files
• Transparent reporting form

### Data availability
This study's data are included in the submitted manuscript and supporting files. Source data have been provided as a compressed directory of supporting tables that correspond to figures as indicated in figure legends. All the scripts for analyzing the RNA-seq datasets and the behavioral assay can be accessed on GitHub, copy archived at swh:1:rev:451bc5d78cd266a00b53612585d201d404f73920. RNA-seq data have been deposited in GEO (accession number: GSE216369).

The following dataset was generated:

| Author(s) | Year | Dataset title | Dataset URL | Database and Identifier |
|---|---|---|---|---|
| Chen J, Costa EK, Singh PP | 2022 | An automated feeding system for the African killifish reveals effects of dietary restriction on lifespan and allows scalable assessment of associative learning | https://www.ncbi.nlm.nih.gov/geo/query/acc.cgi?acc=GSE216369 | NCBI Gene Expression Omnibus, GSE216369 |

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
