## [Editor Report]

The manuscript by McKay et al., describes in detail a novel feeding system for killifish that allows high precision control of feeding amount and schedule on a per-tank basis. The system is open-source, utilizing 3D-printed and off-the-shelf components and due to this emphasis on open software and hardware, the system can be built by individual research groups and the approach therefore appears highly scalable. The authors demonstrate the value of precise control of food intake by investigating sex-specific lifespan effects and associated transcriptome responses to DR in killifish. The resources described in this paper will be a major step in killifish husbandry and management to facilitate its use as a model of longevity studies.

---

## [Decision Letter]

**Decision letter after peer review:**

Thank you for submitting your article "An automated feeding system for the African killifish reveals effects of dietary restriction on lifespan and allows scalable assessment of associative learning" for consideration by *eLife*. Your article has been reviewed by 3 peer reviewers, including Jan Gruber as Reviewing Editor and Reviewer #1, and the evaluation has been overseen by Matt Kaeberlein as the Senior Editor. The following individual involved in review of your submission has agreed to reveal their identity: Ajay Muthuru (Reviewer #3).

Essential revisions:

The reviewers were generally enthusiastic about the design and implementation of the novel feeding system and excited about its potential applications and value to the community. However, the reviewers also agreed that for publication in *eLife* it would be important to provide broader interest beyond the killifish community – e.g. in the form of further data on at least one of the applications introduced. The consensus is that at its current stage, the biological applications are essentially validation experiments, limiting the general interest of the manuscript. While these validation experiments are well done and convincing and while the manuscript details an exciting new methodology, the study requires additional demonstration of the kind of insights that can be generated using the system.

One option would be to focus further on the associative learning results, for example exploring mutants that might extend the length of associative learning and to identify genes / pathways involved. An alternative option would be to further explore of the sex-specific effects of DR on lifespan (e.g. explore reason for the sex specificity of these benefits). Either the DR or the associative learning aspects would need to be significantly strengthened.

---

## [Author Response]

Essential revisions:The reviewers were generally enthusiastic about the design and implementation of the novel feeding system and excited about its potential applications and value to the community. However, the reviewers also agreed that for publication in eLife it would be important to provide broader interest beyond the killifish community – e.g. in the form of further data on at least one of the applications introduced. The consensus is that at its current stage, the biological applications are essentially validation experiments, limiting the general interest of the manuscript. While these validation experiments are well done and convincing and while the manuscript details an exciting new methodology, the study requires additional demonstration of the kind of insights that can be generated using the system.One option would be to focus further on the associative learning results, for example exploring mutants that might extend the length of associative learning and to identify genes / pathways involved. An alternative option would be to further explore of the sex-specific effects of DR on lifespan (e.g. explore reason for the sex specificity of these benefits). Either the DR or the associative learning aspects would need to be significantly strengthened.

We thank the Editorial team and the Reviewers for their enthusiasm for our manuscript and for their very helpful comments and suggestions.

1) A main point brought by the Editors and Reviewers is that it would be important to provide broader interest beyond the killifish community in the form of further data, and an option that was suggested was to explore of the sex-specific effects of dietary restriction (DR) on lifespan. This is a great suggestion. To address this main point, we have now used the automated feeders to identify the molecular pathways associated with the sex-differences in the longevity response of males and females to DR. Specifically:

– We have now generated transcriptomic datasets for 2 key organs (liver and brain) from male and female killifish fed an ad libitum (AL) and DR regimen for 5 weeks, using the automated feeders (new Figure 5A). Analysis of these RNA-seq datasets showed that the liver (but not the brain) exhibited strong transcriptional sexual dimorphism already in the AL conditions, and that this sexual dimorphism was even further enhanced in the DR condition (new Figure 5B, new Figure 5—figure supplement 1A-C).

– Interestingly, we find that DR leads to sexually dimorphic changes in the liver in genes and pathways involved in inflammation, lipid metabolism, and endoplasmic reticulum (ER) homeostasis (new Figure 5C, D, new Figure 5—figure supplement 1D). These sexspecific changes triggered by DR in the liver could underlie at least part of the different longevity response to DR in males and females and some of them are also observed in mammals. The identification of specific genes with strong sex-specificity in response to DR could extend to other species.

Intriguingly, we find that sexually dimorphic genes are enriched among the genes responding to DR (new Figure 5E, new Figure 5—figure supplement 1E). This enrichment points to an interesting relationship between sex and diet, with DR preferentially modulating sexually dimorphic genes.

Thus, sex-specific transcriptomic responses may contribute to the different lifespan response to DR in killifish (a differential that is also observed in other species). We believe that these new transcriptomic studies broaden the scope of our manuscript beyond the killifish, because they propose molecular pathways that are sexually dimorphic and particularly affected by DR which could apply to other species.

2) The Reviewers also had several methodological and technical comments that needed be addressed, particularly on behavioral analyses. To address these, we have bolstered our behavioral analysis by generating a new automated pipeline and benchmarking it on thorough manual analysis. Specifically, we have (1) generated a robust pipeline for unbiased quantification of positive association behaviors by measuring the fish velocity in successive trials (new Figure 6B-E, new Figure 6—figure supplement 1A,B); (2) calculated different automated metrics of success for associative behavior and validated them by manual scoring (new Figure 6F-L, new Figure 6—figure supplement 1C-F); (3) better defined how we calculated a “learning index”; (4) plotted the learning index as a continuum of different ages and sexes, in addition to binning (Figure 6M); (5) performed power analyses to determine the optimal number of animal to use for well-powered behavior experiments. We have also addressed all remaining points by changes in the figures and/or text. Together, we believe that these new experiments and analyses have significantly enhanced our manuscript and broadened the scope of our study. We thank the Editorial team and the Reviewers for suggesting them.